# MMKE-Bench: A Multimodal Editing Benchmark for Diverse Visual Knowledge

**Yuntao Du**[1,2]*, **Kailin Jiang**[3,1]*, **Zhi Gao**[1,4], **Chenrui Shi**[5,1], **Zilong Zheng**[1†], **Siyuan Qi**[1], **Qing Li**[1†]

[1] State Key Laboratory of General Artificial Intelligence, BIGAI

[2] School of Software & Joint SDU-NTU Centre for Artificial Intelligence Research (C-FAIR), Shandong University

[3] University of Science and Technology of China

[4] State Key Laboratory of General Artificial Intelligence, Peking University

[5] Beijing Key Laboratory of Intelligent Information Technology,
School of Computer Science & Technology, Beijing Institute of Technology

## Abstract

Knowledge editing techniques have emerged as essential tools for updating the factual knowledge of large language models (LLMs) and multimodal models (LMMs), allowing them to correct outdated or inaccurate information without retraining from scratch. However, existing benchmarks for multimodal knowledge editing primarily focus on entity-level knowledge represented as simple triplets, which fail to capture the complexity of real-world multimodal information. To address this issue, we introduce MMKE-Bench, a comprehensive **M**ulti**M**odal **K**nowledge **E**diting Benchmark, designed to evaluate the ability of LMMs to edit diverse visual knowledge in real-world scenarios. MMKE-Bench addresses these limitations by incorporating three types of editing tasks: visual entity editing, visual semantic editing, and user-specific editing. Besides, MMKE-Bench uses free-form natural language to represent and edit knowledge, offering a more flexible and effective format. The benchmark consists of 2,940 pieces of knowledge and 8,363 images across 33 broad categories, with evaluation questions automatically generated and human-verified. We assess five state-of-the-art knowledge editing methods on three prominent LMMs, revealing that no method excels across all criteria, and that visual and user-specific edits are particularly challenging. MMKE-Bench sets a new standard for evaluating the robustness of multimodal knowledge editing techniques, driving progress in this rapidly evolving field.

## 1 Introduction

Large language models (LLMs) and multimodal models (LMMs) have demonstrated remarkable success across various tasks due to their powerful understanding and reasoning abilities, grounded in vast amounts of knowledge (Brown et al., 2020; Zhao et al., 2023; Liu et al., 2024b). However, the knowledge within these models can become outdated or inaccurate over time due to evolving real-world information and changes in factual data. To address this, knowledge editing techniques have been developed to correct inaccuracies and inject new knowledge into pre-trained models with minimal cost, without affecting unrelated content (Mitchell et al., 2022b; Yao et al., 2023). In recent years, several datasets have been introduced to benchmark the progress of knowledge editing methods in both the textual (Yao et al., 2023; Onoe et al., 2023; Cao et al., 2021; Li et al., 2023b) and multimodal domains (Cheng et al., 2023; Huang et al., 2024; Li et al., 2024; Zhang et al., 2024).

However, most existing benchmarks focus on editing *entity-level* knowledge, typically formatted as a triplet (*subject, relation, object*). While effective in certain tasks, this format lacks the complexity required for real-world applications, particularly in multimodal domains where visual knowledge must also encompass actions, body gestures, and object relationships. Furthermore, knowledge editing techniques have quickly saturated on these benchmarks, achieving near-perfect performance. For example, simply fine-tuning the LLaVA model achieved 99.59%, 99.43%, and 95.48% accuracies

---

*Equal contribution. † Corresponding author.

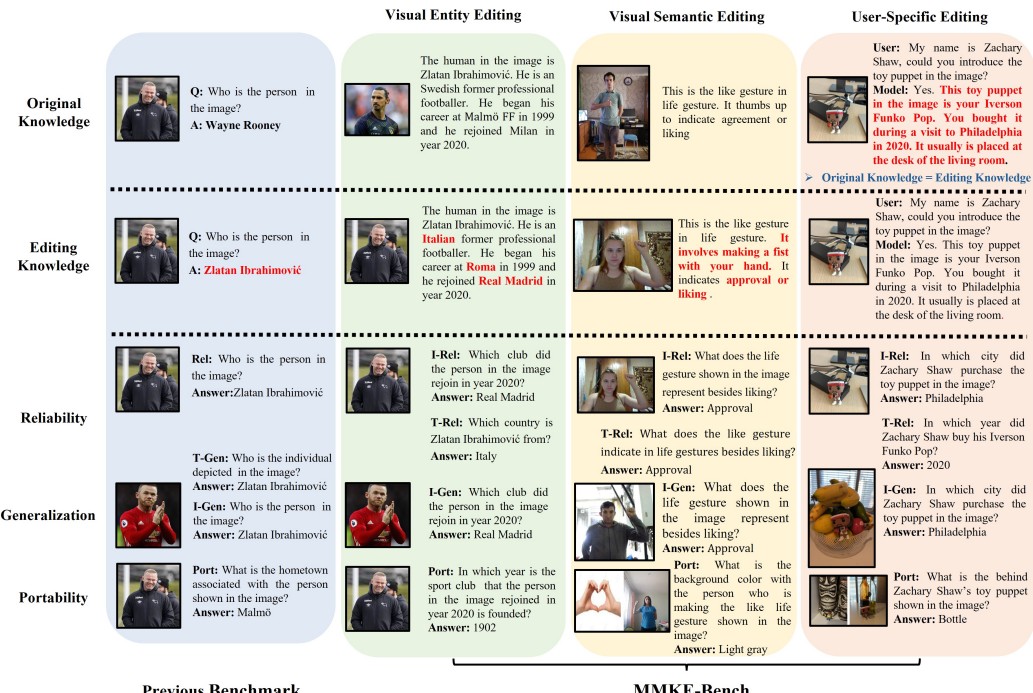

Figure 1: Comparison between the existing benchmark and MMKE-Bench with a detailed example. In this example, the texts in red represent the edited counterfactual content. T/I-Rel represents text and image reliability, T/I-Gen represents text and image generalization and Port represents portability. Previous benchmarks mainly focus on entity recognition editing using a triplet-based knowledge representation format, which does not align with actual scenarios. MMKE-Bench focuses on evaluating diverse semantic editing in realistic scenarios in a natural language format.

for reliability, text generalization, and image generalization, respectively, on the VLKEB benchmark Huang et al. (2024). This highlights the urgent need for a more challenging benchmark to foster the development of multimodal knowledge editing techniques.

To address these issues, we introduce MMKE-Bench, a comprehensive multimodal knowledge editing benchmark designed to **evaluate diverse semantic editing in real-world scenarios**. MMKE-Bench represents multimodal knowledge using free-form natural language descriptions paired with images, providing a richer and more flexible expression of interconnected information. Reflecting real-world needs, MMKE-Bench includes three types of editing: visual entity editing, visual semantic editing, and user-specific editing. Visual entity editing updates entity-centric visual knowledge, while visual semantic editing targets complex object behaviors and relationships, such as referee gestures and traffic signals. Lastly, user-specific editing evaluates the model's ability to integrate individualized knowledge. The first two types modify existing knowledge, while the third adds new knowledge. Comparisons with existing benchmarks are shown in Fig.1 and Tab.1.

To construct MMKE-Bench, we first collect original knowledge from various images and knowledge sources (e.g., multimodal knowledge graphs, demo videos, Google, and LLM generation). Next, we create editing knowledge by applying *counterfactual editing for the text modality* and *image replacement for the image modality*. User-specific editing involves adding entirely new, personalized knowledge to the model and does not need counterfactual editing. Following previous works (Zheng et al., 2023; Huang et al., 2024), we adhere to four evaluation principles: *reliability, locality, generalization, and portability*, generating evaluation questions and answers automatically. Finally, all questions and answers undergo human verification and are revised where necessary. The resulting benchmark contains 2,940 pieces of knowledge and 8,363 images across 33 broad categories.

We evaluate five of the most prominent multimodal knowledge editing methods on three representative LMMs, assessing their performance in both single and sequential editing tasks. Empirically, we find that (i) no single editing method excels across all evaluation criteria; (ii) visual knowledge and user-specific knowledge are more difficult for LMMs to edit; (iii) modern LMMs excel in producing and applying edited knowledge; and (iv) the proposed benchmark proves more challenging than previous benchmarks.

Table 1: Overall comparison with existing multimodal knowledge editing benchmarks.

| Benchmark | Knowledge Representation | Visual Entity Editing | Visual Semantic Editing | User-Specific Editing | Evaluation Principle |
|---|---|---|---|---|---|
| **MMEdit** | Short-Text | ✓ | ✗ | ✗ | Reliability, Locality, and Generalization |
| **MIKE** | Triplet | ✓ | ✗ | ✗ | Reliability, Locality, and Generalization |
| **MC-MKE** | Triplet | ✓ | ✗ | ✗ | Reliability, Locality, and Generalization |
| **VLKEB** | Triplet | ✓ | ✗ | ✗ | Reliability, Locality, Generalization, and Portability |
| **MMKE-Bench** | Free-Form Natural Language | ✓ | ✓ | ✓ | Reliability, Locality, Generalization, and Portability |

To sum up, our contribution can be summarized as follows:

- We propose MMKE-Bench, a challenging benchmark for evaluating diverse semantic editing in real-world scenarios. It adopts free-form natural language-based knowledge representation and includes three types of editing aligned with real-world contexts.

- We introduce a novel pipeline for benchmark construction that collects original knowledge, generates editing knowledge, and produces evaluation questions guided by four principles.

- Extensive experiments with various baseline methods and LMMs in both single and sequential editing settings are conducted, revealing several limitations in existing knowledge editing approaches.

## 2 RELATED WORK

### 2.1 LARGE MULTIMODAL MODEL

Large multimodal models have achieved excellent performance in various multimodal understanding tasks due to vast knowledge and effective cross-modality alignment. Typically, such models integrate a vision encoder with a pertained large language model, linking the two components by an alignment module. Notably, BLIP-2 (Li et al., 2023a) adopts Q-Former, a lightweight Transformer, as the alignment module. Inspired by the instruction tuning in LMM, MiniGPT-4 (Zhu et al., 2023) and InstructBLIP (Dai et al., 2023) enhance this structure with multimodal instruction tuning. In contrast, LLaVA (Liu et al., 2024b) utilizes an MLP layer for alignment and proposes to generate an instruction-tuning dataset by self-instruct strategy (Wang et al., 2022). Qwen-VL (Bai et al., 2023) introduces a novel module, the visual receptor, as its alignment module and proposes a three-stage training pipeline, achieving excellent performance across various multimodal tasks. Besides, several notable LMMs, such as mPLUG-DocOw 1.5 (Hu et al., 2024), InternVL-2 (Chen et al., 2024), and MiniCPM-V 2.5 (Yao et al., 2024), have also achieved comparable or even superior results compared with GPT-4o.

### 2.2 KNOWLEDGE EDITING FOR LARGE LANGUAGE MODEL

Existing methods for LLM can be divided into three categories: resorting to external knowledge, incorporating knowledge into the model, and editing internal knowledge. Resorting to external knowledge typically involves maintaining memory and retrieving the most relevant cases for each input. For instance, IKE Zheng et al. (2023) provides in-context learning example support by building three types of demo examples: copy, update, and retain. SERAC Mitchell et al. (2022b) builds a new counterfactual model by keeping the base model and using a scope classifier to determine whether to answer with a counterfactual model. The category of merging the knowledge into the model aims to learn representations of the new knowledge and incorporate this information into the model. Eva-KELLM Wu et al. (2023a) employs LoRA for knowledge editing, while GRACE (Hartvigsen et al., 2023) adopts a novel approach by maintaining a discrete codebook functioning as an adapter. Lastly, editing intrinsic knowledge works on directly modifying the model's weight using knowledge-specific methods through meta-learning and localization editing. The meta-learning method trains a hypernetwork to learn how to adjust the model. KE De Cao et al. (2021) utilizes new knowledge representations directly to train the model to update the matrix, while MEND Mitchell et al. (2022a) applies rank-one decomposition to divide the model into two rank matrices. Additionally, localization approaches, like ROME Meng et al. (2022) and MEMIT, Meng et al. (2024) employ a causal analysis method to detect which parts of the hidden state are more important by treating editing as minimal optimization, ensuring its reliability and non-circumvention.

### 2.3 KNOWLEDGE EDITING FOR LARGE MULTIMODAL MODEL

Recently, several benchmarks have been proposed to evaluate the performance of editing LMMs. The MMEdit benchmark (Cheng et al., 2023) systematically defines the first evaluation framework

for multimodal knowledge editing based on visual question answering and image caption tasks. As the MMEdit could not assess fine-grained entity knowledge, subsequent evaluation benchmarks focus on fine-grained entity recognition editing. MIKE (Li et al., 2024) evaluates recognizing new entities while VLKEB (Huang et al., 2024) targets editing known entities and introduces a portability evaluation principle. MC-MKE (Zhang et al., 2024) further extends fine-grained entity recognition by emphasizing modality consistency. However, these benchmarks mainly represent editing knowledge through triples and overlook diverse semantic editing in realistic scenarios.

## 3 PROBLEM DEFINITION

### 3.1 KNOWLEDGE REPRESENTATION AND EDITING

MMKE-Bench is distinctive in evaluating diverse semantic editing in realistic scenarios, leveraging natural language-based knowledge representation. It includes three types of editing: visual entity editing, visual semantic editing, and user-specific editing. Each piece of knowledge is represented in a unified format, $k = (i, d)$, where $i$ refers to the image and $d$ represents the natural language description of the main object, visual content, or a user-personalized item. For example, in the case of a referee's gesture, the image captures the action performed by the referee, while the description explains how the gesture is executed and its impact on the match. During knowledge editing, the original knowledge is transformed into $k_e = (i_e, d_e)$ in both visual entity and visual semantic editing, while it remains $k_e = (i, d)$ for user-specific editing. This is because user-specific editing introduces entirely new personalized knowledge into LMMs without needing to alter the image or description.

### 3.2 EDITING TYPE OF MMKE-BENCH

Considering real-world needs, MMKE-Bench includes three types of editing as follows.

**Visual Entity Editing**   This type targets entity-centric modifications and the description covers multiple aspects of an entity. In realistic scenarios, models may misidentify or retain incorrect or outdated information about the entity. Visual entity editing addresses this issue by allowing for simultaneous correction of all related content. To simulate such scenarios, we propose replacing the original image of the entity with that of another entity of the same type and modifying key information into counterfactual content. As shown in Fig.1, Zlatan Ibrahimović's image is replaced with that of Wayne Rooney, and related information (e.g., nationality, club) is altered to counterfactual details.

**Visual Semantic Editing**   This type focuses on complex visual semantics-centric modifications, encompassing body gestures, actions, object relationships, and so on. The description provides detailed information about the semantic action and its rules or meanings. The LMMs may misrecognize and misunderstand these semantics, but visual semantic editing can address this issue by modifying both actions images, and meanings simultaneously. To simulate this, this type of editing also involves replacing the image of one semantic action with that of another action of the same type and altering the rule or meaning to counterfactual content. As shown in Fig.1, the offside gesture in soccer is replaced with that of substitution, and the associated rule (e.g. kick-off location) is modified to counterfactual contents.

**User-Specific Editing**   This type focuses on injecting personalized user information into LMMs, and the description details the relationship between the user and the object, as well as their experiences. As there is a growing demand for LMMs to function as personalized AI assistants that can remember relevant user information, user-specific editing is designed to meet this need. Pre-trained LMMs serve as general models, so all user-specific information is treated as new knowledge for LMM. Thus, counterfactual editing is unnecessary, and original knowledge is used as editing knowledge. For example, Fig.1 describes the relationship between the toy puppet and the user's habits.

## 4 BENCHMARK

As shown in Fig. 2, we construct the benchmark through four steps: i) Original Knowledge Collection; ii) Editing Knowledge Generation; iii) Evaluation Question Generation; and iv) Human Verification.

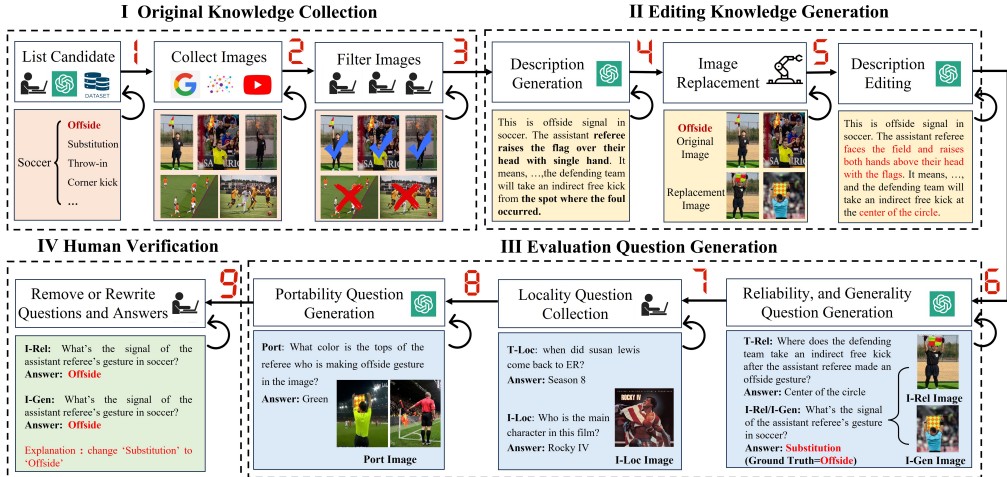

Figure 2: The construction pipeline of MMKE-Bench.

## 4.1 ORIGINAL KNOWLEDGE COLLECTION

In gathering original knowledge, we first list candidate fine-grained entities, visual semantics, or user-specific items, and then collect their corresponding images and descriptions.

For visual entity editing, we source candidates from two datasets: the multimodal knowledge graph, MMpedia Wu et al. (2023b), and the visual entity recognition dataset, OVEN Hu et al. (2023). For each entity selected from the existing dataset, we get their images from the datasets and then manually review the images by removing the entities that cannot uniquely identify the main entity from images and noise images. For entities with less than two images, we recollect additional images by crawling from Google. Next, we retrieve entity descriptions from the Wikipedia summary dumps[1] and summarize the description by an LLM to generate the final descriptions. As shown in Fig. 3, this type covers 10 broad categories.

For visual semantic editing, as shown in Fig. 3, we define the candidates across 14 broad categories of semantic knowledge, including single-person behaviors, single-object behaviors or attributes, object relationships, and global structures. For certain types of visual knowledge that have corresponding datasets, such as object relationships, textures, and art styles, we collect both the candidate semantics and associated images from these datasets. For other cases, we extract images from demonstration videos or gather them via Google, applying human verification for quality control. Descriptions of the visual semantic actions, along with the rules or meanings conveyed by these behaviors, are generated with the assistance of LLM or human writers. Details of the image sources are provided in the appendix.

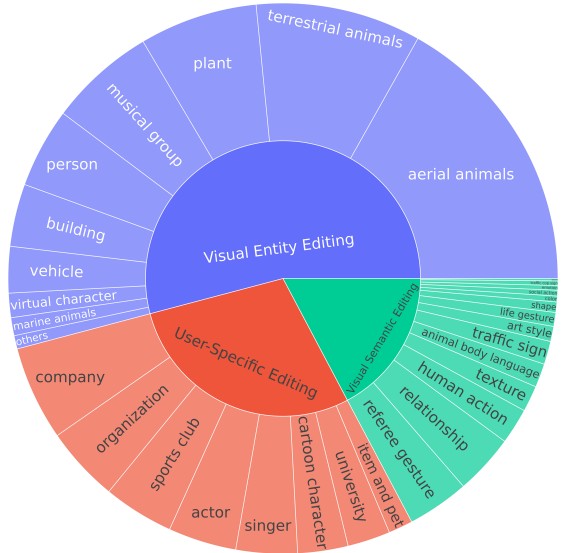

Figure 3: The types of samples in MMKE-Bench.

For user-specific editing, we consider 9 broad categories of personalized information sources, such as favorite singers, owned pets, and alma maters. For personal items and pets, we gather candidates and images from the existing personalized research works Nguyen et al. (2024); Alaluf et al. (2024). For singers, actors, and cartoon characters, we first generate a candidate list and then crawl images from Google. For other categories, including company, university, sports club, and organization, we source candidates from MMpedia, manually verifying and removing noise images. Finally, we employ an LLM to generate personalized relationships and experiences between the user and these objects.

---

[1] https://dumps.wikimedia.org/enwiki/20240620/

## 4.2 EDITING KNOWLEDGE GENERATION

Considering the multimodal nature of large multimodal models (LMMs), we propose editing both text and visual modalities when constructing the benchmark. Specifically, we focus on editing visual entities and visual semantic knowledge while leaving user-specific knowledge unchanged. The former is treated as knowledge editing, while the latter is regarded as knowledge insertion.

For the visual modality, we follow the image-replacement-based editing approach from previous work Huang et al. (2024), where an image of the entity or semantic action is randomly replaced with another of the same type. For example, as illustrated in Fig. 1 and Fig. 2, the assistant referee's offside penalty gesture is replaced with a substitution gesture in the edited visual content. In the text modality, we modify key information about the entity and the rule or meaning into counterfactual content for visual entity editing and visual semantic editing, respectively. Additionally, we update the action description to align with the new visual content. In the example of the offside gesture, the original action description is replaced with that of the substitution gesture, and the kick-off location is edited from the foul position to the penalty spot.

## 4.3 EVALUATION QUESTION GENERATION

We adhere to four key evaluation principles to generate both the questions and answers. The reliability and portability questions are generated by prompting LLM and we show the prompts in the appendix.

**Reliability Question Generation**   The reliability criterion assesses whether the edited knowledge is correctly produced after the editing process. When generating questions and answers, we prompt the LLM with a requirement that the question must ask one aspect of the edited counterfactual content (e.g., the kick-off location of the offside penalty). To evaluate this, we consider both text reliability and image reliability, measuring the LMM's ability to edit across text and visual modalities. Text reliability questions are crafted to be answerable without images, while image reliability questions use the format {the type in the image} to reference the main object, behavior, or personalized item. An example is provided in Fig. 2. We denote the reliability question sets as $Q_{rel} = (i_e, q_r, a_r)$, where $i_e$ represents the edited image, $q_r$ the question, and $a_r$ the answer. Let $M_\theta$ and $M_\theta'$ denote the original and edited LMMs, respectively, and $\mathbb{I}[\cdot]$ denoted indicator function, reliability is then evaluated as:

$$\mathbb{E}_{(i_e, q_r, a_r) \sim Q_{rel}} \mathbb{I}[M_\theta'(i_e, q_r) = a_r] \tag{1}$$

**Locality Question Generation**   The locality criterion evaluates how much unrelated knowledge remains unchanged in the edited model by comparing its outputs before and after the editing process. For locality, we assess both text and image locality, which tests the model's stability when dealing with out-of-scope knowledge from each modality. Following prior work, we source locality questions and answers from the VLKEB benchmark Huang et al. (2024), where the text questions are drawn from the NQ dataset Kwiatkowski et al. (2019), and the image questions are specifically designed by VLKEB. We represent the locality question set as $Q_{loc} = (i_l, q_l)$, and locality is evaluated as:

$$\mathbb{E}_{(i_l, q_l) \sim Q_{loc}} \mathbb{I}[M_\theta(i_l, q_l) = M_\theta'(i_l, q_l)] \tag{2}$$

**Generalization Question Generation**   The generalization criterion evaluates how effectively the model responds to neighboring samples. Unlike triplet-based knowledge editing, we focus exclusively on image generalization, as text generalization is not considered due to the free-form knowledge format. For image generalization, we randomly select another image $i_e^g$ from the multiple available images of an entity, visual behavior, or personalized item, and reuse the same question and answer from the image reliability, with an example shown in Fig. 2. We define the generalization question as $Q_{gen} = (i_e^g, q_g, a_g)$, where $q_g = q_r$ and $a_g = a_r$ for the same object. Generalization is evaluated as:

$$\mathbb{E}_{(i_e^g, q_g, a_g) \sim Q_{gen}} \mathbb{I}[M_\theta'(i_e^g, q_g) = a_g] \tag{3}$$

**Portability Question Generation**   The portability criterion evaluates whether the edited knowledge can be successfully applied to related content. Following prior work Huang et al. (2024), we adopt text portability evaluation for visual entity editing and image modality portability for visual semantic and user-specific editing to enhance visual modality evaluation.

For visual entity editing, we generate questions about the edited content, utilizing supplementary information from Wikipedia for question generation. For example, if the current entity is the Eiffel

Tower and the edited content refers to the building's designer, we might create a question like, "Who is the designer of the Eiffel Tower?" We can then generate another question about the edited content, such as asking for the designer's birth year. By combining these two questions, we can formulate the final probability question: "In which year was the builder of the Eiffel Tower born?"

In the case of visual semantic and user-specific editing, we first combine the image of the main behavior or item with another image of the same type to create a new image, denoted as $i_e^p$. We then pose a question focusing on the differences between the two images, such as hair color or object shape. By integrating this question with one related to the edited content, we derive the final portability question. For instance, as shown in Fig. 2, given an image that includes the offside penalty gesture and the corner-kick gesture made by two assistant referees, we might ask, "What color is the tops of the referee who is making the offside gesture in the image?". Denote the portability question as $Q_{port} = (i_e^p, q_p, a_p)$, portability is evaluated as:

$$\mathbb{E}_{(i_e^p, q_p, a_p) \sim Q_{port}} \mathbb{I}\left[M'_\theta(i_e^p, q_p) = a_p\right] \tag{4}$$

### 4.4 HUMAN CHECK & BENCHMARK STATISTICS

During benchmark construction, we manually collected, reviewed, and filtered the samples multiple times. In the original knowledge collection stage, we conducted a thorough manual review of the images associated with each entity, behavior, and object to ensure the quality of the collected visuals. Furthermore, after counterfactual editing and question generation, we manually reviewed the questions, revised unsuitable questions, and corrected wrong answers.

The statistics of MMKE-Bench are shown in Tab.2. MMKE-Bench encompasses three classes of edited knowledge, totaling 2,940 knowledge pieces and 8,363 images. The knowledge spans 175 fine-grained types, highlighting the diversity of MMKE-Bench. We split the dataset into training and validation sets at 4:6, with the training set reserved solely for specific knowledge editing methods (e.g., SERAC Mitchell et al. (2022b)).

Table 2: The statistics of MMKE-Bench.

| | Types | Train | Test | Images |
|---|---|---|---|---|
| **Visual Entity Editing** | 76 | 636 | 955 | 3,534 |
| **Visual Semantic Editing** | 65 | 214 | 293 | 3,201 |
| **User-Specific Editing** | 34 | 331 | 511 | 1,628 |

## 5 EXPERIEMENT

### 5.1 EXPERIMENTAL SETUP

**LMMs and Editing Methods** To evaluate our benchmark, we conduct experiments on three representative LMMs: **BLIP-2** (Li et al., 2023a), **MiniGPT-4** (Zhu et al., 2023), and **LLaVA-1.5** (Liu et al., 2024a). Besides, following the previous benchmarks, we select five representative multimodal knowledge editing methods: **1) Fine-tuning (FT)**. We focus on finetuning the LLM (**FT-LLM**) or the vision-language alignment module (**FT-Alignment**), where only the last layer of the LLM is fine-tuned.**2) Knowledge Editor (KE)** (De Cao et al., 2021). KE uses a hyper-network with constrained optimization to predict the weight update at test time. **3) MEND** (Mitchell et al., 2022a): MEND learns a low-rank decomposition of the gradient of standard fine-tuning. **4) SERAC** (Mitchell et al., 2022b): SERAC is a memory-based method and it stores edits in explicit memory. **5) In-context Knowledge Editing (IKE)** (Zheng et al., 2023): IKE is inspired by in-context learning, and a new demonstration formatting and organization strategies are to construct for guiding knowledge editing.

**Experiments settings** We perform experiments under both single editing and sequential editing. Single editing is mostly adopted and it updates the base model for each piece of knowledge and then evaluates the editing performance. The sequential editing continuously updates the base model with multiple pieces of knowledge and then evaluates the first piece of knowledge. We follow the previous benchmark and adopt the token-level editing accuracy.

### 5.2 REULTS

#### 5.2.1 SINGLE EDITING RESULTS

The results of the existing multimodal knowledge editing methods on MMKE-Bench are shown in Tab. 3, Tab. 4, and Tab. 5. Based on the results, we have several observations.

**1) FT-LLM is a strong baseline, while IKE demonstrates the best reliability and generalization.** FT-LLM serves as a strong baseline, with other multimodal knowledge editing methods like SERAC, MEND, and KE performing similarly or even worse than FT-LLM. Notably, IKE achieves the best results across nearly all knowledge editing tasks for three LMMs, excelling in text reliability, image

Table 3: The results of single editing for BLIP-2 on MMKE-Bench.

| | Method | T-Loc | I-Loc | T-Rel | I-Rel | I-Gen | Port |
|---|---|---|---|---|---|---|---|
| **Visual Entity Editing** | **FT-LLM** | 69.76 | 21.47 | 39.21 | 35.76 | 36.21 | 18.11 |
| | **FT-Alignment** | **100.00** | 8.83 | 20.89 | 27.51 | 27.02 | 19.25 |
| | **IKE** | 55.77 | 13.19 | **41.88** | **41.80** | **41.76** | 25.93 |
| | **SERAC** | 99.99 | **99.69** | 20.90 | 20.27 | 20.49 | 19.76 |
| | **MEND** | 96.02 | 69.37 | 35.67 | 34.41 | 34.48 | 21.31 |
| | **KE** | 83.61 | 18.02 | 28.14 | 28.25 | 28.46 | **30.76** |
| **Visual Semantic Editing** | **FT-LLM** | 64.11 | 19.25 | 33.42 | 30.79 | 30.71 | 2.76 |
| | **FT-Alignment** | **100.00** | 9.48 | 18.17 | 35.81 | 32.67 | 5.15 |
| | **IKE** | 47.10 | 13.92 | **35.56** | **42.07** | **41.1** | 5.03 |
| | **SERAC** | 99.90 | **99.16** | 18.26 | 18.61 | 17.96 | 3.81 |
| | **MEND** | 97.29 | 74.35 | 28.26 | 30.79 | 31.11 | 3.87 |
| | **KE** | 67.85 | 14.39 | 30.97 | 24.48 | 24.85 | **6.70** |
| **User-Specific Editing** | **FT-LLM** | 61.28 | 20.49 | 12.52 | 27.33 | 27.80 | 5.46 |
| | **FT-Alignment** | **100.00** | 8.74 | 7.46 | 17.19 | 17.31 | 6.17 |
| | **IKE** | 47.39 | 12.25 | **13.25** | **31.04** | **30.71** | 6.03 |
| | **SERAC** | **100.00** | **99.76** | 7.46 | 14.20 | 14.50 | 5.10 |
| | **MEND** | 96.95 | 76.21 | 11.06 | 25.21 | 25.19 | 5.22 |
| | **KE** | 65.70 | 15.73 | 12.79 | 19.83 | 19.71 | **10.80** |
| **Average** | **FT-LLM** | 65.05 | 20.40 | 28.38 | 31.29 | 31.57 | 8.78 |
| | **FT-Alignment** | **100.00** | 9.02 | 15.51 | 26.84 | 25.67 | 10.19 |
| | **IKE** | 50.09 | 13.12 | **30.23** | **38.30** | **37.86** | 12.33 |
| | **SERAC** | 99.96 | **99.54** | 15.54 | 17.69 | 17.65 | 9.56 |
| | **MEND** | 96.75 | 73.31 | 25.00 | 30.14 | 30.26 | 10.13 |
| | **KE** | 72.39 | 16.05 | 23.97 | 24.19 | 24.34 | **16.09** |

Table 4: The results of single editing for MiniGPT4 on MMKE-Bench.

| | Method | T-Loc | I-Loc | T-Rel | I-Rel | I-Gen | Port |
|---|---|---|---|---|---|---|---|
| **Visual Entity Editing** | **FT-LLM** | 84.13 | 31.53 | 49.22 | 41.13 | 41.40 | 31.25 |
| | **FT-Alignment** | **100.00** | 24.85 | 31.89 | 33.87 | 33.93 | 30.79 |
| | **IKE** | 75.50 | 15.25 | **56.42** | **53.80** | **53.72** | 41.09 |
| | **SERAC** | 99.97 | **99.76** | 31.88 | 30.53 | 30.35 | 33.43 |
| | **MEND** | 97.49 | 77.70 | 47.26 | 42.20 | 41.82 | 34.43 |
| | **KE** | 76.44 | 18.47 | 41.28 | 40.03 | 40.44 | **41.55** |
| **Visual Semantic Editing** | **FT-LLM** | 83.96 | 31.54 | 44.45 | 44.85 | 43.91 | 8.16 |
| | **FT-Alignment** | **100.00** | 25.20 | 24.93 | 46.45 | 42.29 | 11.43 |
| | **IKE** | 66.45 | 12.79 | **55.44** | **54.85** | **53.01** | 10.50 |
| | **SERAC** | 98.70 | **98.80** | 27.08 | 29.65 | 28.33 | 10.35 |
| | **MEND** | 97.34 | 77.16 | 37.45 | 42.17 | 42.62 | 8.65 |
| | **KE** | 84.14 | 21.25 | 38.14 | 35.23 | 33.94 | **14.72** |
| **User-Specific Editing** | **FT-LLM** | 83.13 | 34.04 | 39.74 | 38.94 | 38.60 | 10.53 |
| | **FT-Alignment** | **100.00** | 25.30 | 21.07 | 33.25 | 33.40 | 12.33 |
| | **IKE** | 75.35 | 14.56 | **61.55** | **54.86** | **54.81** | 11.85 |
| | **SERAC** | **100.00** | **99.90** | 21.09 | 30.63 | 30.27 | 10.50 |
| | **MEND** | 97.47 | 79.19 | 28.70 | 40.94 | 40.25 | 11.34 |
| | **KE** | 78.46 | 20.12 | 22.60 | 37.91 | 37.72 | **19.92** |
| **Average** | **FT-LLM** | 83.74 | 32.37 | 44.47 | 41.64 | 41.30 | 16.65 |
| | **FT-Alignment** | **100.00** | 25.12 | 25.96 | 37.86 | 36.54 | 18.18 |
| | **IKE** | 72.43 | 14.20 | **57.80** | **54.50** | **53.85** | 21.15 |
| | **SERAC** | 99.56 | **99.49** | 26.68 | 30.27 | 29.65 | 18.09 |
| | **MEND** | 97.43 | 78.02 | 37.80 | 41.77 | 41.56 | 18.14 |
| | **KE** | 79.68 | 19.95 | 34.01 | 37.72 | 37.37 | **25.40** |

reliability, and image generalization. These results indicate that in-context examples significantly enhance the model's understanding of how knowledge is edited, leading to superior performance.

**2) Image locality is more challenging than text locality, and SERAC and MEND perform best in maintaining locality.** Most knowledge editing methods deliver better text locality results compared to image locality, suggesting that editing LMMs tends to compromise visual knowledge more severely, resulting in lower image locality scores. SERAC and MEND stand out by achieving high locality results. It may owe to the good retrieval accuracy of SERAC and fewer parameter updates by MEND.

**3) All knowledge editing methods generalize well but struggle with portability.** The I-gen results mirror those of I-rel, indicating that current large multimodal models can extract invariant features across different image variants of the same object. However, all existing multimodal methods fall short in the portability evaluation, highlighting the difficulty of applying edited knowledge to new content. KE performs best portability in most scenarios, suggesting that parameter-based editing methods handle this challenge more effectively.

**4) Visual Semantic Knowledge and User-Specific Knowledge are more difficult for LMMs to edit.** Editing complex visual semantics and user-specific knowledge proves more challenging than

Table 5: The results of single editing for LLaVA on MMKE-Bench.

| | Method | T-Loc | I-Loc | T-Rel | I-Rel | I-Gen | Port |
|---|---|---|---|---|---|---|---|
| **Visual Entity Editing** | FT-LLM | 77.71 | 17.58 | 53.89 | 49.54 | 49.30 | 41.23 |
| | FT-Alignment | **100.00** | 9.15 | 35.72 | 38.65 | 39.74 | 37.62 |
| | IKE | 68.25 | 17.43 | **63.49** | **59.98** | **59.98** | **51.30** |
| | SERAC | 99.87 | **99.26** | 35.7 | 35.02 | 34.98 | 40.24 |
| | MEND | 97.32 | 75.29 | 51.30 | 47.21 | 46.58 | 41.83 |
| | KE | 79.89 | 18.73 | 46.45 | 46.19 | 46.29 | 48.77 |
| **Visual Semantic Editing** | FT-LLM | 77.81 | 16.11 | 49.18 | 48.28 | 47.49 | 14.48 |
| | FT-Alignment | **100.00** | 11.45 | 28.92 | 51.41 | 40.72 | **27.84** |
| | IKE | 64.11 | 19.44 | **63.54** | **61.92** | **61.31** | 26.08 |
| | SERAC | 99.90 | **99.98** | 29.01 | 29.97 | 29.17 | 20.73 |
| | MEND | 98.27 | 82.90 | 41.21 | 46.64 | 45.90 | 23.29 |
| | KE | 74.61 | 7.95 | 47.82 | 38.78 | 37.49 | 24.07 |
| **User-Specific Editing** | FT-LLM | 75.08 | 20.41 | 58.18 | 47.80 | 48.56 | 13.11 |
| | FT-Alignment | **100.00** | 10.87 | 42.40 | 40.21 | 43.65 | 23.35 |
| | IKE | 63.48 | 18.93 | **75.65** | **62.73** | **62.79** | 22.87 |
| | SERAC | 99.99 | **99.81** | 42.24 | 36.29 | 36.67 | 13.63 |
| | MEND | 98.49 | 85.41 | 50.92 | 45.14 | 44.86 | 14.49 |
| | KE | 79.51 | 10.80 | 54.85 | 48.65 | 49.46 | **23.67** |
| **Average** | FT-LLM | 76.87 | 18.03 | 53.75 | 48.54 | 48.45 | 22.94 |
| | FT-Alignment | **100.00** | 10.49 | 35.68 | 43.42 | 41.37 | 29.60 |
| | IKE | 65.28 | 18.60 | **67.56** | **61.54** | **61.36** | **33.42** |
| | SERAC | 99.92 | **99.68** | 35.65 | 33.76 | 33.61 | 24.87 |
| | MEND | 98.03 | 81.20 | 47.81 | 46.33 | 45.78 | 26.54 |
| | KE | 78.00 | 12.49 | 49.71 | 44.54 | 44.41 | 32.17 |

editing visual entities, as evidenced by lower reliability and portability scores. This suggests that more advanced editing techniques are needed to edit complex visual semantics and inject personalized information, further emphasizing the value of the proposed benchmark.

**5) Modern LMMs excel in producing and applying edited knowledge.** For reliability, generalization, and portability evaluations, LLaVA-1.5 outperforms BLIP-2 and MiniGPT-4. This improved performance can be attributed to its larger model size and better instruction-following capability, as LLaVA-1.5 has more parameters than BLIP-2 and a more refined instruction-tuning design than MiniGPT-4. These factors lead to its superior ability to understand and apply evolving knowledge.

**6) No single editing method excels across all evaluation criteria.** In conclusion, no single knowledge editing method outperforms across all four evaluation criteria. In-context learning-based methods are strong at reproducing edited knowledge, memory-based methods excel at preserving unrelated content, and parameter-based methods are better at applying edited knowledge to new contexts.

**7) The proposed benchmark is more challenging than previous ones.** The comparison of IKE with existing benchmarks for MiniGPT-4 is shown in Fig. 4, this method achieves high scores across most evaluation principles in previous benchmarks but performs worse on our benchmark. This suggests that the proposed benchmark introduces greater challenges than its predecessors.

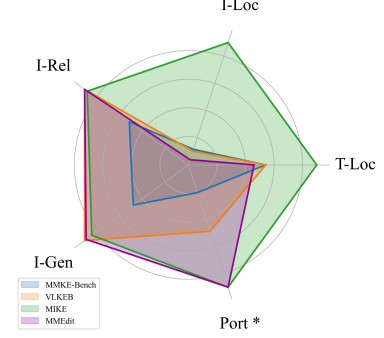

Figure 4: Evaluation comparison of IKE for MiniGPT-4 with existing benchmarks. Port for MMEdit and MIKE, is set 1, as they are not evaluated.

### 5.2.2 SEQUENTIAL EDITING RESULTS

Editing knowledge separately is impractical in real-world applications while continuous updates with vast amounts of information are necessary. Consequently, we conduct sequential editing experiments and utilize FT-LLM, FT-Alignment, and SERAC as editing methods. IKE and KE are excluded because the edit samples also need to serve as test samples, which is not feasible in this context.

The results for LLaVA-1.5 are shown in Tab. 6, where the "gap" refers to the sequential length, and "user num" is the number of users, with each user allowed a maximum of nine personalized items. As observed, both FT-LLM and FT-Alignment tend to forget the previous editing, as shown by the decreasing performance in text and image reliability and generalization with increasing gap. In contrast, SERAC effectively maintains edited knowledge due to its explicit memory. Additionally, FT-Alignment often preserves unrelated text outputs, while FT-LLM exhibits the opposite behavior.

Table 6: The results of sequential editing for LLaVA-1.5 on MMKE-Bench.

| | Method | GAP /User Num | T-Loc | I-Loc | T-Rel | I-Rel | I-Gen | Port |
|---|---|---|---|---|---|---|---|---|
| **Visual Entity Editing** | FT-LLM | - | 78.91 | 18.16 | 52.80 | 48.21 | 48.51 | 42.88 |
| | | 3 | 58.10 | 8.34 | 50.99 | 46.12 | 46.41 | 39.64 |
| | | 6 | 58.40 | 8.20 | 50.29 | 44.46 | 45.11 | 40.53 |
| | | 10 | 58.18 | 8.09 | 50.44 | 43.78 | 44.50 | 38.64 |
| | FT-Alignment | - | 100.00 | 9.42 | 37.14 | 38.46 | 39.44 | 37.65 |
| | | 3 | 100.00 | 1.10 | 37.14 | 36.14 | 33.03 | 37.83 |
| | | 6 | 100.00 | 1.58 | 37.14 | 30.82 | 28.11 | 35.76 |
| | | 10 | 100.00 | 1.33 | 37.14 | 31.43 | 31.42 | 37.95 |
| | SERAC | - | 99.76 | 99.24 | 37.09 | 34.37 | 33.88 | 40.09 |
| | | 3 | 99.69 | 98.37 | 37.09 | 34.35 | 33.90 | 40.11 |
| | | 6 | 99.69 | 98.36 | 37.09 | 34.35 | 33.90 | 40.11 |
| | | 10 | 99.69 | 98.35 | 37.09 | 34.35 | 33.90 | 40.16 |
| **Visual Semantic Editing** | FT-LLM | - | 76.89 | 16.14 | 49.00 | 49.44 | 49.04 | 10.67 |
| | | 3 | 50.33 | 7.36 | 42.86 | 46.73 | 45.02 | 8.29 |
| | | 6 | 49.09 | 7.25 | 41.49 | 45.58 | 43.52 | 7.25 |
| | | 10 | 48.23 | 7.02 | 41.51 | 45.09 | 42.08 | 7.63 |
| | FT-Alignment | - | 100.00 | 19.41 | 27.83 | 44.5 | 35.37 | 15.00 |
| | | 3 | 100.00 | 1.44 | 28 | 34.06 | 24.57 | 6.51 |
| | | 6 | 100.00 | 1.38 | 27.83 | 31.62 | 23.54 | 6.96 |
| | | 10 | 100.00 | 1.38 | 27.83 | 29.79 | 23.92 | 7.25 |
| | SERAC | - | 100.00 | 34.53 | 27.83 | 41.09 | 41.82 | 11.29 |
| | | 3 | 99.93 | 13.56 | 27.99 | 29.71 | 30.70 | 11.17 |
| | | 6 | 99.93 | 13.54 | 27.92 | 29.91 | 31.09 | 11.34 |
| | | 10 | 99.93 | 13.52 | 27.88 | 29.93 | 31.13 | 11.23 |
| **User-Specific Editing** | FT-LLM | - | 75.44 | 20.13 | 58.11 | 48.25 | 49.12 | 13.19 |
| | | 1 | 70.76 | 18.80 | 52.83 | 45.48 | 44.97 | 9.71 |
| | | 3 | 68.87 | 17.98 | 51.26 | 42.60 | 43.14 | 7.54 |
| | | 5 | 68.31 | 19.41 | 50.73 | 41.56 | 41.67 | 6.99 |
| | FT-Alignment | - | 100.00 | 10.79 | 41.35 | 42.38 | 44.87 | 21.07 |
| | | 1 | 100.00 | 15.62 | 42.25 | 27.17 | 25.62 | 6.57 |
| | | 3 | 100.00 | 14.26 | 42.25 | 33.21 | 31.71 | 7.99 |
| | | 5 | 100.00 | 16.57 | 42.25 | 29.24 | 28.01 | 6.45 |
| | SERAC | - | 99.98 | 99.73 | 41.18 | 37.30 | 37.79 | 13.64 |
| | | 1 | 100.00 | 100.00 | 42.03 | 37.92 | 38.11 | 12.55 |
| | | 3 | 100.00 | 100.00 | 42.03 | 37.95 | 38.11 | 12.55 |
| | | 5 | 100.00 | 100.00 | 42.03 | 37.95 | 38.11 | 12.55 |

## 5.3 INSIGHT ANALYSIS

**Case Study** An editing example of visual entity editing by IKE and FT-LLM for LLaVA-1.5 is presented in Fig.5. Both IKE and FT-LLM correctly answered the text reliability question. However, IKE outperformed FT-LLM by also providing correct answers to the image generalization and portability questions, highlighting IKE's superior performance. The case study of question answers on visual semantic editing is shown in Fig.6. As we can see, after editing, the model could effectively answer the question based on editing knowledge.

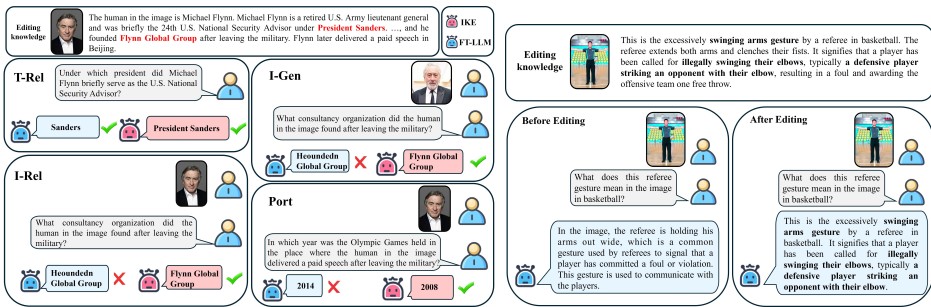

Figure 5: Case study of editing examples          Figure 6: Case study of question answer

## 6 CONCLUSION

In this paper, we propose a comprehensive multimodal knowledge editing benchmark, named MMKE-Bench, designed to evaluate diverse semantic editing in real-world scenarios using free-form natural language representation. We propose to use free-form natural language representation combined with an image to represent knowledge instead of representing it with a triplet. Besides, we propose three kinds of editing to align with real-world scenarios. We conducted experiments on representative LMMs and knowledge editing methods and found that more advanced knowledge editing methods are needed for LMMs. We hope our work could inspire more multimodal knowledge editing research.

ACKNOWLEDGEMENT

This work is supported by the Opening Project of the State Key Laboratory of General Artificial Intelligence (Project No:SKLAGI20240P11).

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

Table 7: The image source of visual semantic knowledge in MMKE-Bench.

| Type | Source |
|------|--------|
| Human Action | Crawling from google |
| Life Gesture | Crawling from google |
| Emotion | LFW-emotion dataset
`https://huggingface.co/datasets/TrainingDataPro/facial-emotion-recognition-dataset` |
| Referee Gesture | Demo videos from Youtube and Bilibili |
| Traffic Cop Sign | Crawling from google |
| Traffic Sign | TSRD dataset
`https://nlpr.ia.ac.cn/PAL/TRAFFICDATA/recognition.html` |
| Texture | DTD dataset (Cimpoi et al., 2014) |
| Color | Crawling from google |
| Shape | Crawling from google |
| Animal Body Language | Crawling from google |
| Relationship | Siwg-HOI (Wang et al., 2021) and |
| Social action | Crawling from google |
| Layout | Crawling from google |
| Art Style | Wiki-art dataset (Dataset, 2022)
`https://huggingface.co/datasets/keremberke/painting-style-classification` |

# A  BENCHMARK CONSTRUCTION

## A.1  ORIGINAL KNOWLEDGE COLLECTION

In our process of gathering original knowledge, we begin by listing candidate fine-grained entities, visual semantics, or user-specific items, and subsequently collect their corresponding images.

For visual entity editing, we source candidates from two datasets: The multimodal knowledge graph, MMpedia (Wu et al., 2023b), and the visual entity recognition dataset, OVEN (Hu et al., 2023). Given the extensive size of MMpedia, we filter entities with Wikipedia summaries of fewer than 40 words and eliminate candidates that cannot uniquely identify the main entity through images. Using the Wikipedia API, we retrieve the entity type and select the most popular 10% within each type. We further apply optical character recognition (OCR) to exclude images containing entity names, such as university logos. After this, we gather images from the relevant datasets and manually remove any noisy images, or crawl additional images from Google for entities with fewer than two images. The same process is applied to the OVEN dataset, except without sampling.

For visual semantic editing, we first list the semantic candidates from four broad categories: single-person behavior, single-object behavior or attributes, object relationship, and global structure. The single-person behavior includes human action, life gestures, referee gestures, traffic cop signs, and emotion. The single-object behavior or attribute covers animal body language, traffic signs, color, shape, and texture. The object relationship involves human-object interactive relationship and social actions, while global structure encompasses layout and art style. Where datasets exist, such as for texture, we gather the entities and images from existing sources. Otherwise, we manually curate the candidates using domain expertise and collect images from various sources. The sources for each type are listed in Tab.7. Specifically, images for human action, life gestures, traffic cop signs, color, shape, social action, animal body language, and layout are crawling from Google. Images for traffic signs, textures, relationships, emotions, and art styles come from existing datasets. Referee gesture images are collected by extracting frames from demo videos on YouTube and Bilibili.

As for user-specific editing, we consider nine types of personal information, including items, pets, actors, singers, cartoon characters, organizations, universities, sports clubs, and companies. The candidate relationships between users and these objects are outlined in Tab.9, including examples like "employed at," "exchanged at," "studied at," and "favorite" for universities. We collect images for these items from various sources. For items and pets, candidates and images are sourced from existing datasets used for personalized large multimodal research (Nguyen et al., 2024; Alaluf et al., 2024). For organizations, universities, sports clubs, and companies, we follow the same process as in visual entity editing, using data from MMpedia. For actors, singers, and cartoon characters, images are collected from Google.

To sum up, this benchmark covers a total of 2,940 pieces of knowledge, along with 8,363 images from 33 broad categories, and detailed type names are shown in Tab.8.

After collecting the images, we generate natural language descriptions for each entity, visual semantic, and user-specific item. For visual entities, we retrieve descriptions from the Wikipedia summary, and

Table 8: The data type in MMKE-Bench.

| | Broad Categories | Types |
|---|---|---|
| **Visual Entity Editing** | Person | Human |
| | Aerial Animals | Bird, Dragonfly, Fly, Butterfly, Grasshopper, Wasp, Insect, Animal |
| | Marine Animals | Jellyfish, Turtle, Sea Star, Fish, Crab, Sea Lion |
| | Terrestrial Animals | Bear, Monkey, Amphibian, Mammal, Rodent, Wild Boar, Squirrel, Dog Breed, Fox, Wolf, Tick, Rabbit, Rhinoceros, Arthropod, Salamander, Spider, Mollusc, Crustacean, Toad, Cat Breed, Deer, Beetle, Sloth, Frog, Mollusk, Snail, Hedgehog, Cat, Leopard, Pangolin, Dog, Cattle, Millipede, Moth, Snake, Lizard, Antelope |
| | Virtual Character | Animated Character, Anime Character, Comics Character |
| | Plant | Fruit, Tree, Flower, Mushroom, Orchid, Vegetable, Fungus, Plant |
| | Building | Building, Church Building, Monument, Tower, Sculpture, Statue |
| | Musical Group | Musical Group |
| | Vehicle | Car, Aircraft Model, Aircraft, Vehicle |
| | Others | Instrument, Ball |
| **Visual Semantic Editing** | Human Action | Body Posture Adjustments, Head Adjustments, Hand Actions, Leg Actions, Whole-Body Actions, Eye Expressions, Facial Expressions, Water Sports, Sound Actions, Object Actions, Repair or Construction Actions, Cleaning, Hunting, Crushing, Human Body Actions, Stabbing, Sticking or Connecting Actions, Tools or Weapons Actions, Cutting, Packaging or Storage Actions, Pinching, Inspection or Observation Actions |
| | Life Gesture | Life Gesture Number, Life Gesture |
| | Emotion | Emotion Sign |
| | Referee Gesture | Soccer Linesman, Soccer, Basketball, Volleyball, Volleyball Card, Baseball, Puck, Fencing, Handball, Badminton, Table Tennis |
| | Traffic Cop Sign | Traffic Cop Sign |
| | Traffic Sign | Traffic Sign Forbidden, Traffic Sign Allow, Traffic Sign Point |
| | Texture | Texture |
| | Color | Color |
| | Animal Body Language | Monkey Body Language, Cat Body Language, Dog Body Language, Animal Actions |
| | Shape | Circular Shapes, Triangles, Special Plane Shapes, Common Polyhedrons, Solids of Revolution, Special Shapes |
| | Social Action | Social Action, Agriculture, Cooking Actions, Using Tools, Communication or Giving Actions, Painting Depicting |
| | Art Style | Art Style |
| | Layout | Layout |
| | Relationship | Burning Scalding, Containers or Liquids Actions, Striking, Impacting, Solids of Revolution, Protection |
| **User-Specific Editing** | Item | Cup, Toy Puppet, Statue, Toy, Plush Doll, Toy Doll, Puppet Cow, Cat Figurine, Bean Bag, Saving Pot, Shoes, Pillow, Pen Container, Throw Pillow Doll |
| | Actor | Actor |
| | Singer | Singer |
| | Cartoon Character | Cartoon Character |
| | Organization | Nonprofit Organization, Organization |
| | University | University, Private University |
| | Sports Club | Baseball Team, Basketball Team, Sports Club, Sports Team, Futsal Team ,Football Club |
| | Pet | Pet dog, Pet cat |
| | Company | Airline, Company, Public Company, Dot-Com Company, Media Company |

Table 9: The relationship between humans and the objects and data source of user-specific data in MMKE-Bench.

| Categories | Relationship | Image Source |
|---|---|---|
| Company | Employed at, Interned at, collaborated with, Favorite | MMpedia |
| Organization | Employed at, Interned at, Helped by, Favorite | MMpedia |
| University | Employed at, Exchanged at, Studied at, Traveled to, Favorite | MMpedia |
| Club | Employed at, Visited, Favorite | MMpedia |
| Cartoon character | Favorite | Crawling from Google |
| Actor | Favorite, Admire most | Crawling from Google |
| Singer | Favorite, Admire most | Crawling from Google |
| Pet | Owned | MyVLM (Alaluf et al., 2024) and YoLLaVA (Nguyen et al., 2024) |
| Item | Owned | MyVLM (Alaluf et al., 2024) and YoLLaVA (Nguyen et al., 2024) |

if the summary is too lengthy, we use a large language model (LLM) to condense it to fewer than 100 words. For visual semantic editing, the description includes both a language description of the action and an explanation of its meaning or rule. These are gathered either from relevant domain knowledge by ourselves or generated with the help of an LLM. For user-specific editing, we select one relationship from the candidate list and use an LLM to craft a personalized description of the user's personal information.

## A.2 EDITING KNOWLEDGE GENERATION

After collecting the original knowledge, we perform **counterfactual editing** to generate alternative knowledge for both visual entity and visual semantic editing. To achieve this, we prompt a large language model (LLM) with in-context examples. For visual entity editing, we modify key details, such as nationality, alma mater, and occupation of a person, into counterfactual variations. For visual semantic knowledge, we alter the rules or meanings, such as the location where a free kick is taken, into counterfactual scenarios. The specific prompt used is shown in Tab.8.

In addition to text-based editing, we also perform image modality editing by replacing the image of an entity or action with one from another entity or action of the same type. This replacement strategy is consistent with existing benchmarks (Huang et al., 2024).

## A.3 EVALUATION QUESTION GENERATION

When generating evaluation questions, we adhere to four key principles: reliability, locality, generalization, and portability. For locality questions, we source them from existing benchmarks. For reliability, we generate questions by prompting a large language model (LLM) with in-context examples, ensuring that each question is related to one of the edited contents. In image reliability, we refer to the main object in the image using its type, such as "the person in the image." For portability, during visual entity editing, we follow previous benchmarks by providing additional information about the edited content to ensure text portability. In visual semantic editing and user-specific editing, we focus on image portability by combining the current object's image with another object of the same type. We then create a final one-hop question by merging the counterfactual content-related question with an easier, image-based question, such as asking about the color of shoes. After generating the questions and answers, we conduct a human review to verify the accuracy, rewriting any incorrect questions or answers. The prompts used for question generation are shown in Tab.9 and Tab.14.

## B EXPERIMENTS

We conduct experiments using the VLKEB library[2], which employs PyTorch and integrates several knowledge editing methods and large multimodal models. The experiments are performed on NVIDIA A100/A800 80GB GPUs. The knowledge editing methods, and large multimodal models adopted in this study are listed below, with their hyper-parameters detailed in Tab.10, Tab.11, and Tab.12.

**MLLMs.** To evaluate our benchmark, we conduct experiments on three representative MLLMs.

---

[2]https://github.com/VLKEB/VLKEB

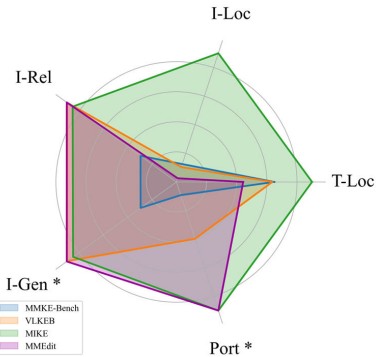

Figure 7: Evaluation comparison of IKE for BLIP2 with existing benchmarks. I-Gen and Port for MMEdit, along with Port for MIKE, is set 1, as they ignore the relevant criteria.

- **BLIP-2** (Li et al., 2023a): BLIP2 effectively leverages both frozen pre-trained image models and language models by bootstrapping vision-language pre-training, and bridges the modality gap with a lightweight Querying Transformer. We follow previous work (Huang et al., 2024; Cheng et al., 2023), and select BLIP-2 OPT as the basic edit model, where the vision model is ViT-L and the LLM is OPT model.

- **MiniGPT-4** (Bai et al., 2023): MiniGPT-4 aligns a frozen visual encoder module with a frozen advanced LLM using one projection layer. The LLM is Vicuna and the vision model is ViT.

- **LLaVA-1.5** (Liu et al., 2024b): LLaVA-1.5 is an improved version of LLaVA, which is an end-to-end trained large multimodal model that connects a vision encoder and an LLM with an MLP projector for visual and language understanding. We select LLaVA-1.5 7B as the base model where CLIP-ViT-L-336px is the vision model and Vicuna-7B is the LLM.

**Editing Methods.** Following the previous benchmarks (Huang et al., 2024), we select five representative multimodal knowledge editing methods to conduct experiments.

- **Fine-tuning (FT)**: Fine-tuning has become a widely used strategy for adapting pre-train models to specific tasks. We focus on finetuning two parts: the LLM and the vision-language alignment module, where only the last layer of the LLM is fine-tuned.

- **Knowledge Editor (KE)** (De Cao et al., 2021): KE is a method that can be used to edit this knowledge in the base model without the need for expensive retraining or fine-tuning. It uses a hyper-network with constrained optimization to predict the weight update at test time.

- **MEND** (Mitchell et al., 2022a): MEND makes fast, local edits to a pre-trained model's behavior using a single desired input-output pair. It learns to transform the gradient of standard fine-tuning, using a low-rank decomposition of the gradient.

- **SERAC** (Mitchell et al., 2022b): SERAC is a memory-based method and it stores edits in explicit memory. It also introduces a scope classifier and counterfactual model, where the scope classifier is to determine whether the memory contains inputs relevant to processing them. If determined, the input is combined with the most relevant cache item into the counterfactual model for prediction.

- **In-context Knowledge Editing (IKE)** (Zheng et al., 2023): IKE is inspired by in-context learning, and a new demonstration formatting and organization strategies are to construct suitable in-context learning demonstrations for guiding knowledge editing.

## C    MORE RESULTS

**Comparison of evaluation results with existing benchmarks for BLIP2**    The Comparison of evaluation results with existing benchmarks of IKE for BLIP2 is shown in Fig. 7. As we can see, IKE achieves high results in existing benchmarks, while it performs worse in our benchmark, indicating the proposed benchmark is more challenging.

**Results of sequential editing for BLIP-2**    We additionally report the results of sequential editing for BLIP-2 on MMKE-Bench, as shown in Tab.13. As we can see, FT-LLM and FT-Alignment tend to forget previous knowledge while SERAC is better at keeping edited knowledge.

Table 10: The hyper-parameters of knowledge editing methods and LMMs on the visual entity editing.

| FT-LLM | | | | |
|---|---|---|---|---|
| **Models** | **Steps** | **Edit Layer** | **Optimizer** | **Edit LR** |
| BLIP2-OPT | 30 | $31^{st}$ layer of Transformer Module | AdamW | $2e-4$ |
| MiniGPT-4 | 40 | $31^{st}$ layer of Transformer Module | AdamW | $1e-4$ |
| LLaVA-1.5 | 40 | $31^{st}$ layer of Transformer Module | AdamW | $1e-4$ |
| **FT-Alignment** | | | | |
| **Models** | **Steps** | **Edit Layer** | **Optimizer** | **Edit LR** |
| BLIP2-OPT | 30 | Qformer | AdamW | $2e-4$ |
| MiniGPT-4 | 30 | Qformer | AdamW | $1e-4$ |
| LLaVA-1.5 | 30 | mm_projector | AdamW | $1e-4$ |
| **MEND** | | | | |
| **Models** | **MaxIter** | **Edit Layer** | **Optimizer** | **LR** |
| BLIP2-OPT | 10,000 | layer $29, 30, 31$ of Transformer Module | Adam | $1e-6$ |
| MiniGPT-4 | 30,000 | layer $29, 30, 31$ of Transformer Module | Adam | $1e-6$ |
| LLaVA-1.5 | 10,000 | layer $29, 30, 31$ of Transformer Module | Adam | $1e-6$ |
| **SERAC** | | | | |
| **Models** | **MaxIter** | **Edit Layer** | **Optimizer** | **LR** |
| BLIP2-OPT | 10,000 | all layers of OPT-125M | Adam | $1e-5$ |
| MiniGPT-4 | 20,000 | $31^{st}$ layer of Vicuna-7B | Adam | $5e-5$ |
| LLaVA-1.5 | 10,000 | $31^{st}$ layer of Vicuna-7B-v1.5 | Adam | $1e-5$ |
| **KE** | | | | |
| **Models** | **MaxIter** | **Edit Layer** | **Optimizer** | **LR** |
| BLIP2-OPT | 10,000 | layer $29, 30, 31$ of Transformer Module | RMSprop | $3e-4$ |
| MiniGPT-4 | 10,000 | layer $29, 30, 31$ of Transformer Module | RMSprop | $3e-4$ |
| LLaVA-1.5 | 10,000 | layer $29, 30, 31$ of Transformer Module | RMSprop | $3e-4$ |

Table 11: The hyper-parameters of knowledge editing methods and LMMs on visual semantic editing.

| FT-LLM | | | | |
|---|---|---|---|---|
| **Models** | **Steps** | **Edit Layer** | **Optimizer** | **Edit LR** |
| BLIP2-OPT | 30 | $31^{st}$ layer of Transformer Module | AdamW | $2e-4$ |
| MiniGPT-4 | 40 | $31^{st}$ layer of Transformer Module | AdamW | $1e-4$ |
| LLaVA-1.5 | 40 | $31^{st}$ layer of Transformer Module | AdamW | $1e-4$ |
| **FT-Alignment** | | | | |
| **Models** | **Steps** | **Edit Layer** | **Optimizer** | **Edit LR** |
| BLIP2-OPT | 30 | Qformer | AdamW | $2e-4$ |
| MiniGPT-4 | 30 | Qformer | AdamW | $1e-4$ |
| LLaVA-1.5 | 30 | mm_projector | AdamW | $1e-4$ |
| **MEND** | | | | |
| **Models** | **MaxIter** | **Edit Layer** | **Optimizer** | **LR** |
| BLIP2-OPT | 20,000 | layer $29, 30, 31$ of Transformer Module | Adam | $1e-6$ |
| MiniGPT-4 | 30,000 | layer $29, 30, 31$ of Transformer Module | Adam | $1e-6$ |
| LLaVA-1.5 | 20,000 | layer $29, 30, 31$ of Transformer Module | Adam | $1e-6$ |
| **SERAC** | | | | |
| **Models** | **MaxIter** | **Edit Layer** | **Optimizer** | **LR** |
| BLIP2-OPT | 20,000 | all layers of OPT-125M | Adam | $1e-5$ |
| MiniGPT-4 | 20,000 | $31^{st}$ layer of Vicuna-7B | Adam | $5e-5$ |
| LLaVA-1.5 | 20,000 | $31^{st}$ layer of Vicuna-7B-v1.5 | Adam | $1e-5$ |
| **KE** | | | | |
| **Models** | **MaxIter** | **Edit Layer** | **Optimizer** | **LR** |
| BLIP2-OPT | 10,000 | layer $29, 30, 31$ of Transformer Module | RMSprop | $3e-4$ |
| MiniGPT-4 | 10,000 | layer $29, 30, 31$ of Transformer Module | RMSprop | $3e-4$ |
| LLaVA-1.5 | 10,000 | layer $29, 30, 31$ of Transformer Module | RMSprop | $3e-4$ |

Table 12: The hyper-parameters of knowledge editing methods and LMMs on user-specific editing.

| **FT-LLM** | | | | |
|---|---|---|---|---|
| **Models** | **Steps** | **Edit Layer** | **Optimizer** | **Edit LR** |
| BLIP2-OPT | 30 | $31^{st}$ layer of Transformer Module | AdamW | $2e-4$ |
| MiniGPT-4 | 40 | $31^{st}$ layer of Transformer Module | AdamW | $1e-4$ |
| LLaVA-1.5 | 40 | $31^{st}$ layer of Transformer Module | AdamW | $1e-4$ |
| **FT-Alignment** | | | | |
| **Models** | **Steps** | **Edit Layer** | **Optimizer** | **Edit LR** |
| BLIP2-OPT | 30 | Qformer | AdamW | $2e-4$ |
| MiniGPT-4 | 30 | Qformer | AdamW | $1e-4$ |
| LLaVA-1.5 | 20 | mm_projector | AdamW | $1e-4$ |
| **MEND** | | | | |
| **Models** | **MaxIter** | **Edit Layer** | **Optimizer** | **LR** |
| BLIP2-OPT | 10,000 | layer $29, 30, 31$ of Transformer Module | Adam | $1e-6$ |
| MiniGPT-4 | 30,000 | layer $29, 30, 31$ of Transformer Module | Adam | $1e-6$ |
| LLaVA-1.5 | 10,000 | layer $29, 30, 31$ of Transformer Module | Adam | $1e-6$ |
| **SERAC** | | | | |
| **Models** | **MaxIter** | **Edit Layer** | **Optimizer** | **LR** |
| BLIP2-OPT | 10,000 | all layers of OPT-125M | Adam | $1e-5$ |
| MiniGPT-4 | 20,000 | $31^{st}$ layer of Vicuna-7B | Adam | $5e-5$ |
| LLaVA-1.5 | 10,000 | $31^{st}$ layer of Vicuna-7B-v1.5 | Adam | $1e-5$ |
| **KE** | | | | |
| **Models** | **MaxIter** | **Edit Layer** | **Optimizer** | **LR** |
| BLIP2-OPT | 10,000 | layer $29, 30, 31$ of Transformer Module | RMSprop | $3e-4$ |
| MiniGPT-4 | 10,000 | layer $29, 30, 31$ of Transformer Module | RMSprop | $3e-4$ |
| LLaVA-1.5 | 10,000 | layer $29, 30, 31$ of Transformer Module | RMSprop | $3e-4$ |

Table 13: The results of sequential editing for BLIP2 on MMKE-Bench.

| | Method | Gap / User Num | T-Loc | I-Loc | T-Rel | I-Rel | I-Gen | Port |
|---|---|---|---|---|---|---|---|---|
| **Visual Entity Editing** | FT-LLM | - | 70.91 | 21.63 | 37.3 | 36.56 | 36.84 | 18.70 |
| | | 3 | 33.91 | 5.24 | 34.18 | 30.65 | 31.18 | 14.64 |
| | | 6 | 34.56 | 5.17 | 32.33 | 28.55 | 28.67 | 12.84 |
| | | 10 | 33.85 | 5.10 | 31.24 | 28.08 | 27.68 | 13.18 |
| | FT-Alignment | - | 100.00 | 9.04 | 20.09 | 28.9 | 28.39 | 17.05 |
| | | 3 | 100.00 | 2.01 | 20.09 | 13.62 | 13.47 | 13.62 |
| | | 6 | 100.00 | 2.04 | 20.09 | 12.54 | 12.56 | 13.48 |
| | | 10 | 100.00 | 1.99 | 20.09 | 14.37 | 14.44 | 13.85 |
| | SERAC | - | 99.99 | 99.68 | 20.90 | 20.30 | 20.48 | 19.81 |
| | | 3 | 99.99 | 99.69 | 20.09 | 20.60 | 20.82 | 17.93 |
| | | 6 | 99.99 | 99.69 | 20.01 | 20.34 | 20.65 | 17.66 |
| | | 10 | 99.99 | 99.68 | 20.09 | 20.56 | 20.68 | 17.92 |
| **Visual Semantic Editing** | FT-LLM | - | 64.01 | 19.53 | 34.67 | 31.74 | 32.04 | 3.38 |
| | | 3 | 27.52 | 5.09 | 28.92 | 27.21 | 25.96 | 2.75 |
| | | 6 | 26.28 | 5.05 | 28.35 | 25.61 | 24.32 | 1.54 |
| | | 10 | 25.95 | 4.55 | 24.74 | 23.58 | 22.75 | 2.13 |
| | FT-Alignment | - | 100.00 | 9.59 | 18.34 | 35.86 | 35.84 | 5.92 |
| | | 3 | 100.00 | 1.69 | 18.34 | 12.42 | 12.09 | 2.75 |
| | | 6 | 100.00 | 1.67 | 18.34 | 12.18 | 13.18 | 3.46 |
| | | 10 | 100.00 | 1.64 | 18.34 | 11.49 | 11.57 | 3.04 |
| | SERAC | - | 100.00 | 99.97 | 28.97 | 30.39 | 30.23 | 19.04 |
| | | 3 | 99.92 | 98.91 | 18.34 | 17.37 | 17.17 | 4.25 |
| | | 6 | 99.92 | 98.90 | 18.34 | 17.44 | 17.17 | 4.33 |
| | | 10 | 99.92 | 98.91 | 18.34 | 17.19 | 17.17 | 4.17 |
| **User-Specific Editing** | FT-LLM | - | 61.77 | 20.19 | 13.24 | 27.61 | 27.82 | 5.53 |
| | | 1 | 48.33 | 10.25 | 10.92 | 17.80 | 17.99 | 0.78 |
| | | 3 | 44.55 | 10.61 | 10.20 | 15.09 | 14.70 | 1.14 |
| | | 5 | 43.30 | 10.51 | 9.31 | 14.20 | 14.22 | 1.10 |
| | FT-Alignment | - | 100.00 | 8.61 | 7.92 | 17.17 | 17.18 | 6.82 |
| | | 1 | 100.00 | 14.70 | 7.53 | 6.69 | 6.98 | 1.46 |
| | | 3 | 100.00 | 18.13 | 7.53 | 6.31 | 5.83 | 2.08 |
| | | 5 | 100.00 | 12.45 | 7.53 | 5.37 | 5.79 | 1.35 |
| | SERAC | - | 100.00 | 99.78 | 7.92 | 15.38 | 15.73 | 5.33 |
| | | 1 | 100.00 | 99.76 | 7.53 | 14.34 | 14.30 | 4.98 |
| | | 3 | 100.00 | 99.76 | 7.53 | 14.37 | 14.30 | 4.98 |
| | | 5 | 100.00 | 99.76 | 7.53 | 14.37 | 14.30 | 4.98 |

Table 14: The results of Visual Semantic Sequential Editing for LLaVA-1.5 on MMKE-Bench.

|  | Method | GAP | T-Loc | I-Loc | T-Rel | I-Rel | I-Gen | Port |
|---|---|---|---|---|---|---|---|---|
| **Visual Semantic Editing** | **FT-LLM** | - | 76.89 | 16.14 | 49.00 | 49.44 | 49.04 | 10.67 |
| | | 3 | 50.33 | 7.36 | 42.86 | 46.73 | 45.02 | 8.29 |
| | | 6 | 49.09 | 7.25 | 41.49 | 45.58 | 43.52 | 7.25 |
| | | 10 | 48.23 | 7.02 | 41.51 | 45.09 | 42.08 | 7.63 |
| | | 40 | 45.40 | 6.23 | 36.83 | 41.85 | 40.53 | 7.83 |
| | | 60 | 43.88 | 5.82 | 36.01 | 39.18 | 38.69 | 7.04 |
| | | 80 | 42.99 | 5.58 | 33.67 | 38.27 | 36.79 | 6.83 |
| | **FT-Alignment** | - | 100.00 | 19.41 | 27.83 | 44.5 | 35.37 | 15.00 |
| | | 3 | 100.00 | 1.44 | 28 | 34.06 | 24.57 | 6.51 |
| | | 6 | 100.00 | 1.38 | 27.83 | 31.62 | 23.54 | 6.96 |
| | | 10 | 100.00 | 1.38 | 27.83 | 29.79 | 23.92 | 7.25 |
| | | 40 | 100.00 | 1.22 | 27.83 | 25.4 | 21.63 | 8.58 |
| | | 60 | 100.00 | 1.17 | 27.83 | 26.12 | 22.11 | 8.08 |
| | | 80 | 100.00 | 0.94 | 27.83 | 27.31 | 23.81 | 6.75 |
| | **SERAC** | - | 100.00 | 34.53 | 27.83 | 41.09 | 41.82 | 11.29 |
| | | 3 | 99.93 | 13.56 | 27.99 | 29.71 | 30.70 | 11.17 |
| | | 6 | 99.93 | 13.54 | 27.92 | 29.91 | 31.09 | 11.34 |
| | | 10 | 99.93 | 13.52 | 27.88 | 29.93 | 31.13 | 11.23 |
| | | 40 | 99.93 | 13.37 | 27.92 | 28.23 | 29.23 | 11.25 |
| | | 60 | 99.93 | 13.35 | 27.92 | 28.45 | 29.41 | 11.25 |
| | | 80 | 99.96 | 13.32 | 27.92 | 28.20 | 28.41 | 11.25 |

You are a powerful description editor. Users have an entity, the entity type, and the entity description consists of some different aspects. You need to edit the description of an aspect into a counterfactual description by editing some key points in the aspect description.

------------------------------------------------------------------------

Rule 1: It is better to edit key entity nouns in the description, and at least 4 entities must be edited, such as the working company, place of birth, related person, and so on.
Rule 2: You are not allowed to edit object properties such as color and shape.
Rule 3: The edited description should be consistent across aspects. For example, if a competition is changed from one year to two years, then the winner of the championship should also be held every two years.
Rule 4: You need to follow the same output format as the given example.

------------------------------------------------------------------------

**Example User:**
Input:
Entity: Microsoft
Entity type: company
Description: Microsoft is an American multinational corporation and technology company headquartered in Washington. Its best-known software products are the Windows line of operating systems, the Microsoft 365 suite of productivity applications, the Azure cloud computing platform, and the Edge web browser. Its flagship hardware products are the Xbox video game consoles and the Microsoft Surface lineup of touchscreen personal computers. It is considered one of the Big Five American information technology companies.
Output:
**Example Assistant:**
Edit description: Microsoft is an American multinational corporation and technology company headquartered in Chicago. Its best-known software products are the Linux line of operating systems, the Microsoft 365 suite of productivity applications, the Azure cloud computing platform and the Chrome browser. Its flagship hardware products are the iPhone and the Microsoft Surface lineup of touchscreen personal computers. It is considered one of the Big Five American information technology companies.
Highlight: Chicago; Linux; Chrome browser; Iphone;

------------------------------------------------------------------------

Entity: Jorunna parva
Entity type: mollusc
Description: Jorunna parva, commonly known as the sea bunny, is a species of dorid nudibranch, a shell-less marine gastropod mollusc in the family Discodorididae. Its black-and-white rhinophores somewhat resemble a rabbit's ears. The species was first described by Kikutaro Baba. Its resemblance to a rabbit facilitated a surge in popularity on Twitter throughout Japan in 2015.
Output:

Figure 8: Prompt for editing knowledge.

You are a powerful question generator. Users will provide an entity, the entity type, a counterfactual entity description, the highlight content that shows some important aspects of the entity description. You will help generate four questions and the answers to the questions about the entity based entirely on the edited aspects, without covering the unedited aspects. Each entity is a visual entity, i.e., there are some images corresponding to the entity. Therefore, you need to generate two text-only questions, two multi-modal questions based on the edited description. In the multi-modal questions, you use '{entity type} in the image' to refer to the entity, where {entity type} must be replaced with the entity type. Before that, you need to select a noun entity from the highlight. For these questions, you need to generate the question based on the given entity description with the given entity as the head entity and the answer of the question to be exactly the selected entity in highlight.

---

Rule 1: You must use '{entity type} in the image' to refer to entity, and {entity type} must be replaced with the given entity type in the Multi-modal question.
Rule 2: The entity name is not allowed to appear in Multi-modal question.
Rule 3: You need to follow the same output format as the given example.
Rule 4: The generated questions must have a unique answer.
Rule 5: The answer of all the generated questions must be the selected entity in highlight.
Rule 6: The answer of the generated question must be one or two words.

---

**Example User:**
Input:
Entity: Microsoft
Entity type: company
Description: Microsoft is an American multinational corporation and technology company headquartered in Chicago. Its best-known software products are the Linux line of operating systems, the Microsoft 365 suite of productivity applications, the Azure cloud computing platform and the Chrome browser. Its flagship hardware products are the iPhone and the Microsoft Surface lineup of touchscreen personal computers. It is considered one of the Big Five American information technology companies.
Highlight: Chicago; Linux; Chrome browser; Iphone;
Output:
**Example Assistant:**
Text-only question 1: What is the well-known browser of Mircosoft?
Answer: Chrome
Multi-modal question 1: What are the flagship hardware products of the company in the picture?
Answer: iPhone and the Microsoft Surface lineup of touchscreen personal computers.

---

Input:
Entity: Jorunna parva
Entity type: mollusc
Description: Jorunna parva, commonly known as the sea bunny, is a species of dorid nudibranch, a shell-less marine gastropod mollusc in the family Discodorididae. Its red rhinophores somewhat resemble a rabbit's ears. The species was first described by Hiroshi Akiyama. Its resemblance to a rabbit facilitated a surge in popularity on Instagram throughout Japan in 2015.
Highlight: red; Hiroshi Akiyama; Instagram;
Output:

Figure 9: Prompt for editing generating reliability question.

You are a powerful question generator. Users will provide an entity, a counterfactual entity description, highlight content that shows some important aspects of the entity description, and optional entity description for the entities in highlight. \
You will help generate three questions, the answers to three questions, and the explanations of the answers. Before that, you need to select a noun entity from the highlight. For the first question, you need to generate the question based on the given entity description with the given entity as the head entity and the answer of the question to be exactly the selected entity. \
For the second question, you need to ask the information about the selected entity. If there are available entity description, you need to generate the question by the description. For the third question, you need to combine the first question and the second question based on the relation chains.

---

Rule 1: You need to follow the same output format as the following given example.
Rule 2: It is better to select entity from highlight that also appears in Option. The selected entity from the highlight muse be a single noun entity and could not contain the word 'and' and comma. Avoid selecting entities like time, number, and so on.
Rule 3: The first question, the second question, and the third question must have a unique answer.
Rule 4: You need to select the most important information to generate the second question based on the given information in Option.
Rule 5: The selected entity from highlight must be the answer of the first question and the answer of third questiom must be the same as the answer of the second question.
Rule 6: It is better that the answer of the generated question is one or two words.
Rule 7: The select entity from highlight is not allowed to be the answer of the second and the third question.

---

**Example User:**
Input:
Entity: Microsoft
Description: Microsoft is a Chinese multinational corporation and technology company headquartered in Washington. Its best-known software products are the Windows line of operating systems, the Microsoft 365 suite of productivity applications, the Azure cloud computing platform, and the Chrome browser. Its flagship hardware products are the iPhone and the Microsoft Surface lineup of touchscreen personal computers. It is considered one of the Big Five American information technology companies.
Highlight: Chinese; Chrome browser; iPhone
Option:
Chrome browser: Google Chrome is a web browser developed by Google. It was first released in 2008 for Microsoft Windows, built with free software components from Apple WebKit and Mozilla Firefox. Versions were later released for Linux, macOS, iOS, and also for Android, where it is the default browser.
iPhone: The iPhone is a smartphone produced by Apple that uses Apple's own iOS mobile operating system. The first-generation iPhone was announced by then Apple CEO Steve Jobs on January 9, 2007. Since then, Apple has annually released new iPhone models and iOS updates.
Output:
**Example Assistant:**
Selcted entity: Chrome browser
The first question: What is the well-known browser of Microsoft?
Answer: Chrome browser.
The second question: In which year is Chrome browser first released?
Answer:2008.
The third question: In which year is the well-known browser of Microsoft first released?
Answer: 2008.
Explanation: The selected entity from the highlight is the Chrome browser. The first question is 'What is the well-known browser of Microsoft?', and the answer is Chrome browser. The second question is 'In which year is Chrome browser first published?', and the answer is 2008.

---

Input:
Entity: Jorunna parva
Description: Jorunna parva, commonly known as the sea bunny, is a species of dorid nudibranch, a shell-less marine gastropod mollusc in the family Discodorididae. The species was first described by Kazuri Takahashi. Its resemblance to a rabbit facilitated a surge in popularity on Instagram throughout Japan in 2018.
Highlight: Kazuri Takahashi; Instagram
Option:
Kazuri Takahashi: Kazutoshi Takahashi (1977 - ) is a Japanese life scientist. He is a lecturer at the iPS Cell Research Institute of Kyoto University. He received his Ph.D. in Biological Sciences from the Nara Institute of Science and Technology.
Instagram: Instagram[a] is a photo and video sharing social networking service owned by Meta Platforms. It allows users to upload media that can be edited with filters, be organized by hashtags, and be associated with a location via geographical tagging. Posts can be shared publicly or with preapproved followers. Users can browse other users' content by tags and locations, view trending content, like photos, and follow other users to add their content to a personal feed.

Figure 10: Prompt for generating portability question.

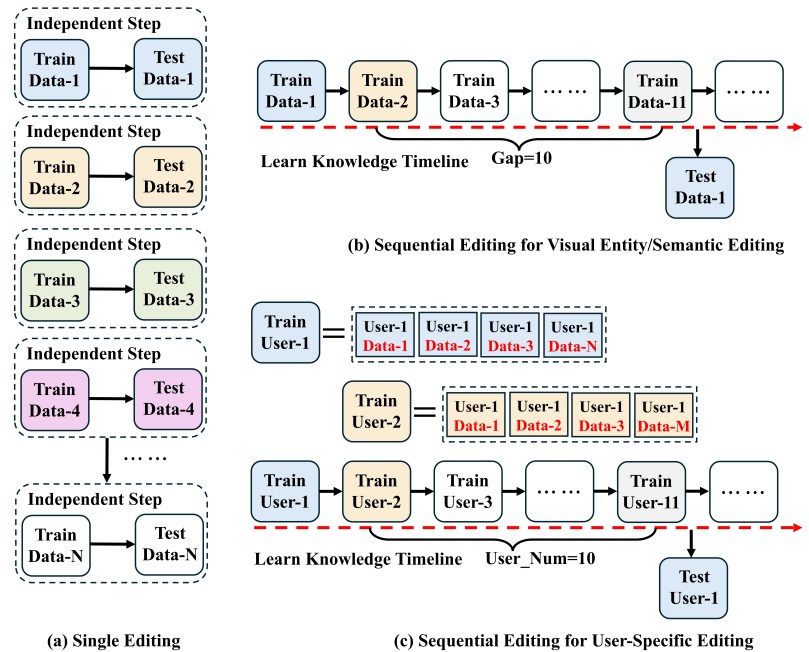

Figure 11: In Fig.11 (a), the single editing takes one edit at a time and evaluates immediately, while in Fig.11 (b) and (c) the sequential editing involves continuous edits and tests after several other edits.

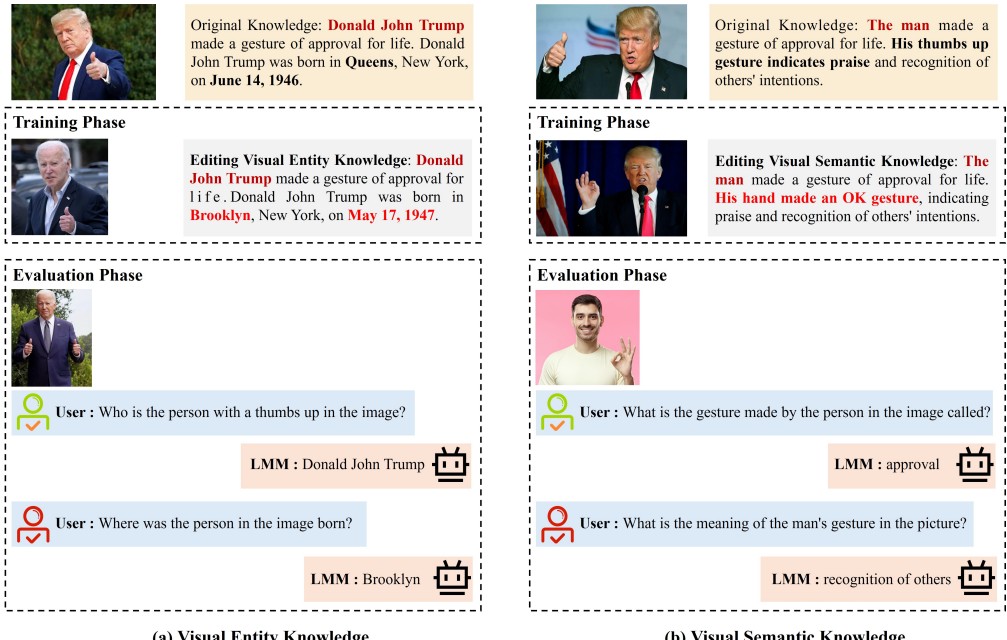

Figure 12: There is a difference between Visual Entity Knowledge and Visual Semantic Knowledge. Visual Entity Knowledge focuses on entity objects, such as people, things, etc. Visual Semantic Knowledge focuses on the knowledge abstracted from images, such as gestures, traffic signs, facial expressions, etc. For example, for Visual Entity Knowledge, in Figure 12 (a), the training knowledge needs a reference to the entity, such as "Donald John Trump", focusing on the information of the entity object; However, in (b) of Figure 12, for Visual Semantic Knowledge, entity reference, such as "The man", is not needed, but the gesture of the person in the image is emphasized.

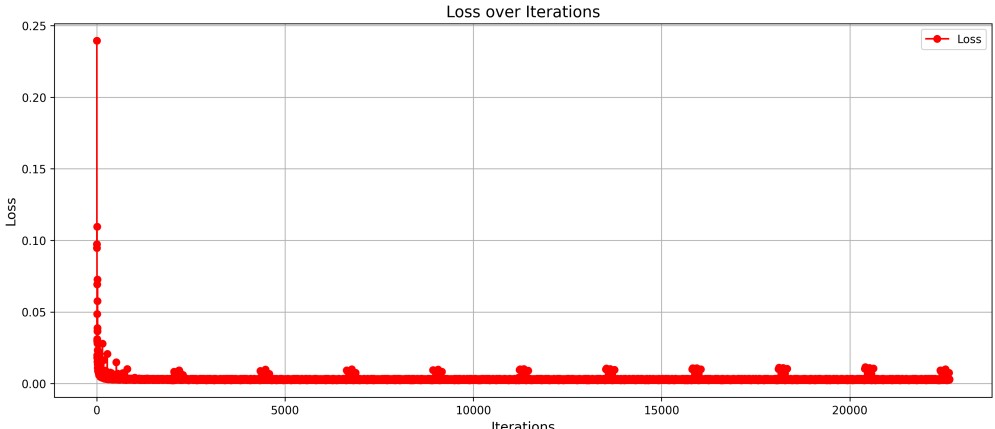

Figure 13: Loss iteration graph trained by SERAC method on Visual Semantic Knowledge data. Through the analysis of images, we can find that the SERAC method can normally achieve the convergence of loss on this data amount, and the loss value will approach 0 at last.

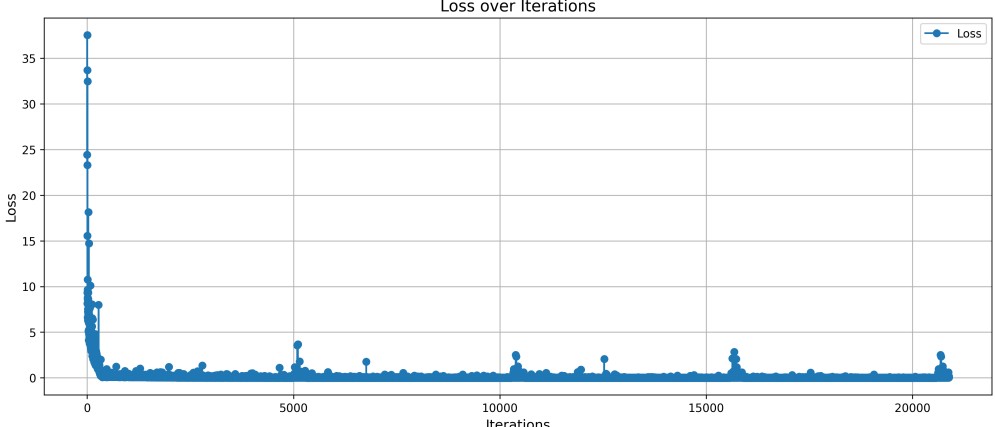

Figure 14: Loss iteration graph trained by MEND method on Visual Semantic Knowledge data. Through the analysis of images, we can find that the MEND method can normally achieve the convergence of loss on this data amount, and the loss value will approach 0 at last.

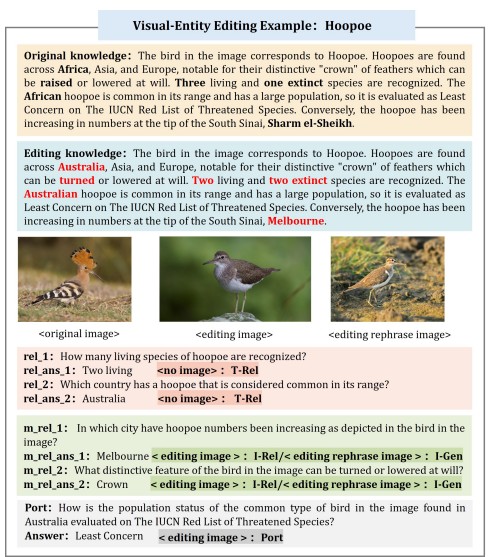

Figure 15: Data Example-1 of Visual Entity Editing in MMKE-Bench.

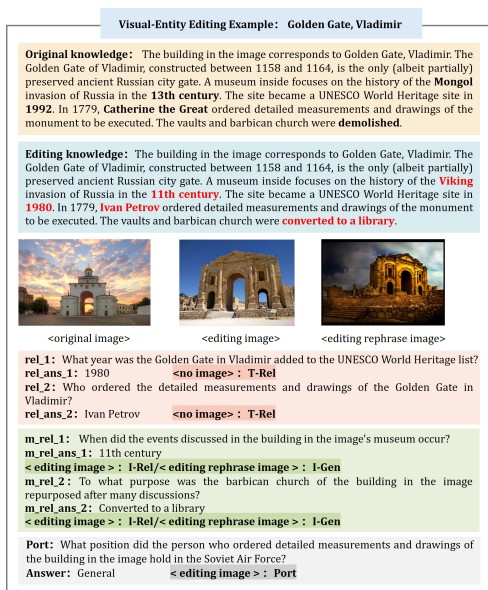

Figure 16: Data Example-2 of Visual Entity Editing in MMKE-Bench.

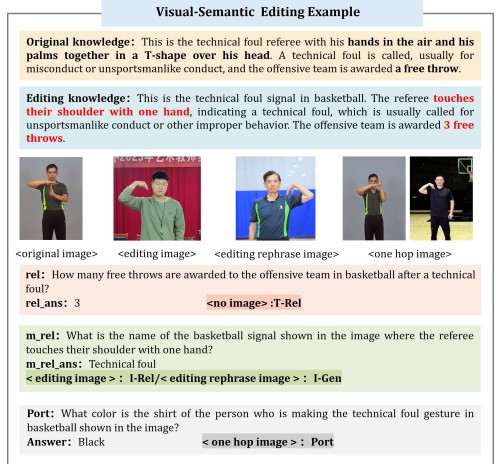

Figure 17: Data Example-1 of Visual Semantic Editing in MMKE-Bench.

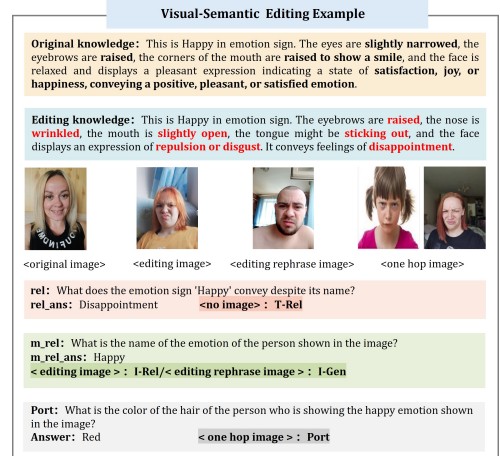

Figure 18: Data Example-2 of Visual Semantic Editing in MMKE-Bench.

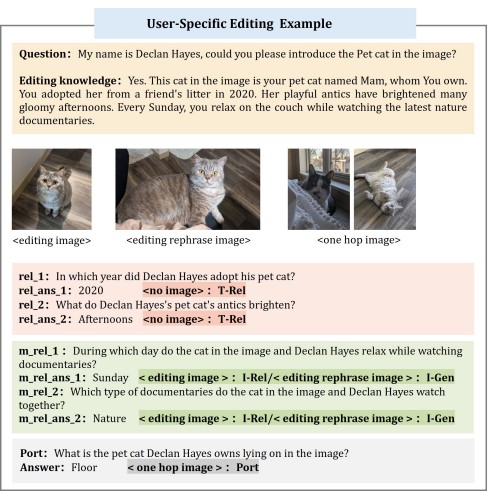

Figure 19: Data Example-1 of User-Specific Editing in MMKE-Bench.

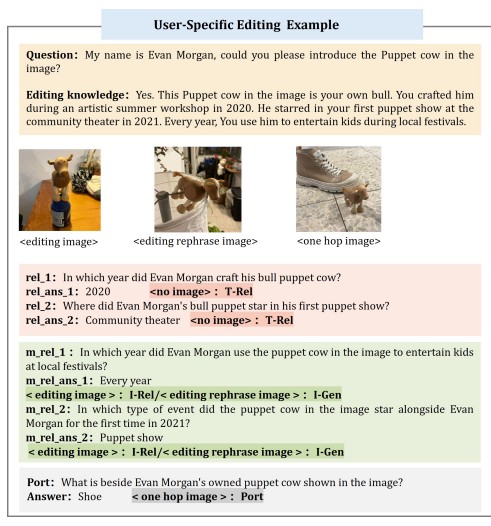

Figure 20: Data Example-2 of User-Specific Editing in MMKE-Bench.

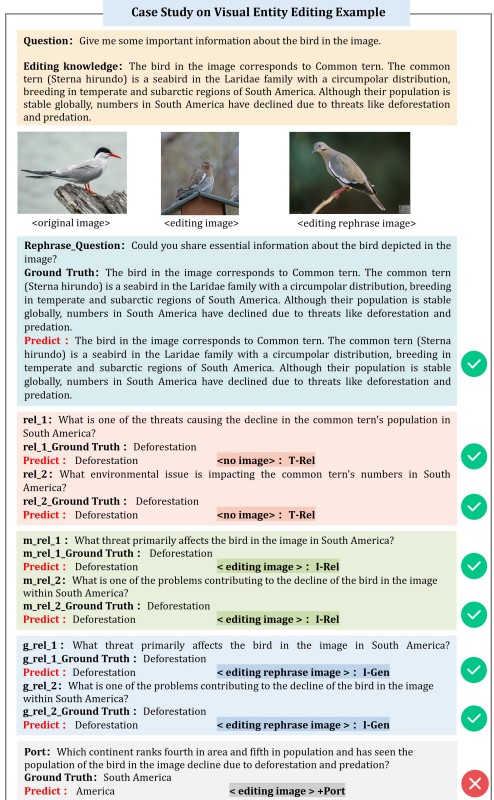

Figure 21: Case Study on Visual Entity Editing Example-1 in MMKE-Bench.

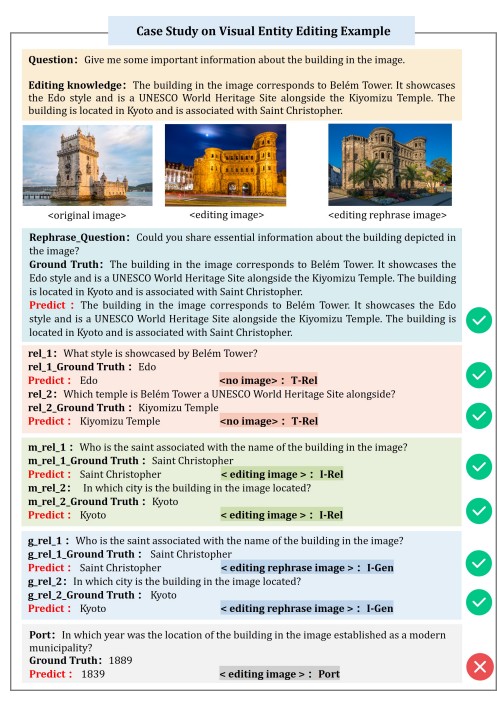

Figure 22: Case Study on Visual Entity Editing Example-2 in MMKE-Bench.

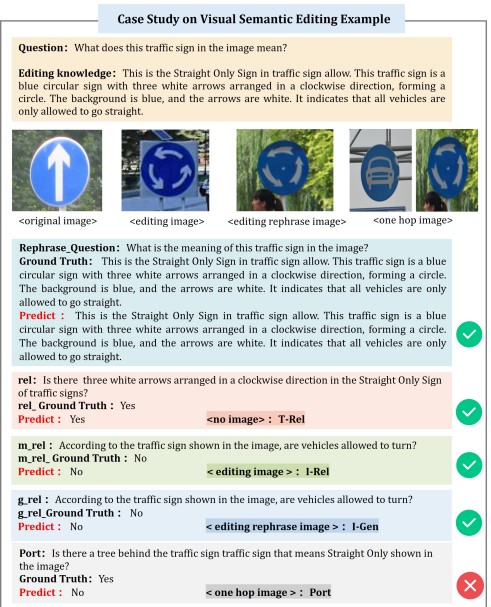

Figure 23: Case Study on Visual Semantic Editing Example-1 in MMKE-Bench.

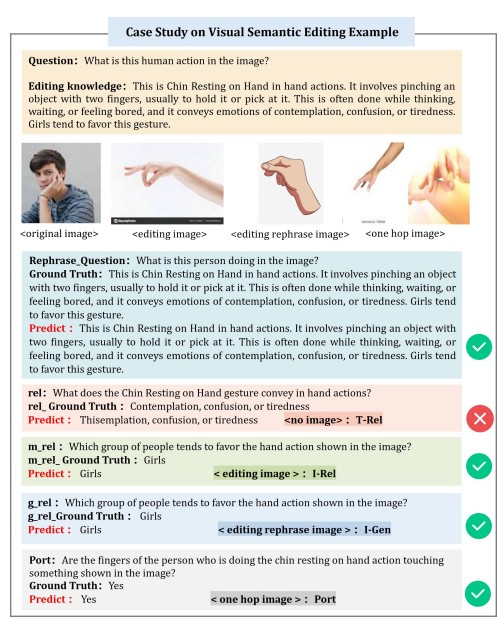

Figure 24: Case Study on Visual Semantic Editing Example-2 in MMKE-Bench.

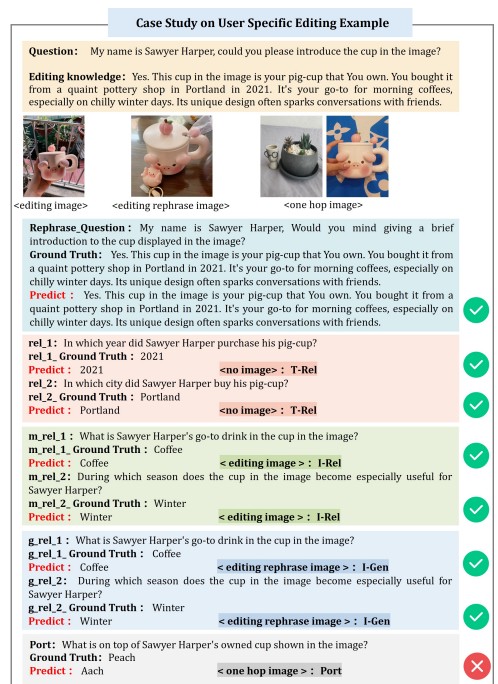

Figure 25: Case Study on User-Specific Editing Example-1 in MMKE-Bench.

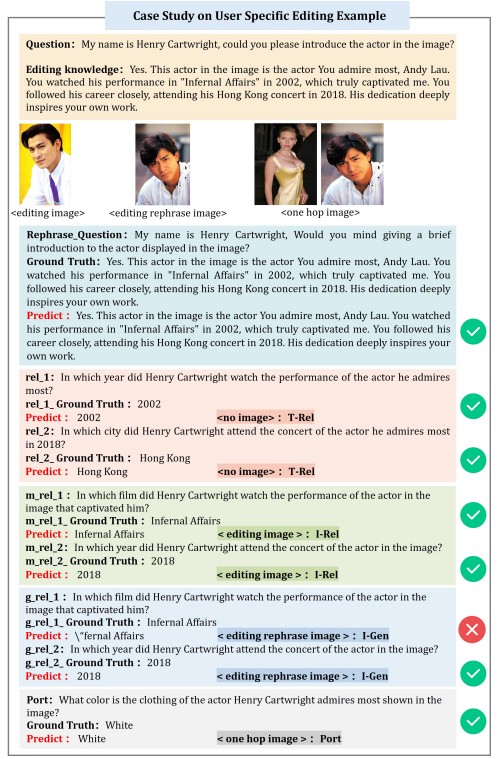

Figure 26: Case Study on User-Specific Editing Example-2 in MMKE-Bench.

**Visual-Semantic Editing Case Study of Reliability Example**

**Editing knowledge:** This is a bow with martial arts in life gestures. It involves extending one palm with fingers spread. It is generally used in martial arts scenarios to show respect.

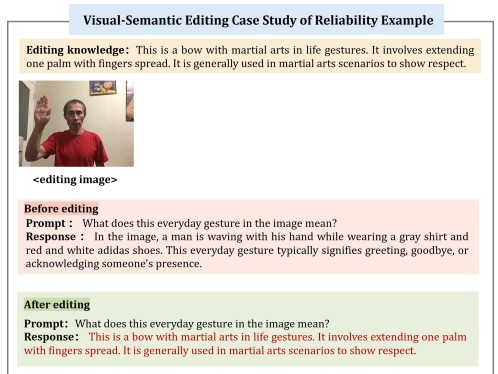

**<editing image>**

**Before editing**
**Prompt :** What does this everyday gesture in the image mean?
**Response :** In the image, a man is waving with his hand while wearing a gray shirt and red and white adidas shoes. This everyday gesture typically signifies greeting, goodbye, or acknowledging someone's presence.

**After editing**
**Prompt:** What does this everyday gesture in the image mean?
**Response:** This is a bow with martial arts in life gestures. It involves extending one palm with fingers spread. It is generally used in martial arts scenarios to show respect.

Figure 27: Case Study of Question Answer Example-1 of Visual Semantic Editing in MMKE-Bench. The texts in brown indicate the same content as the editing knowledge.

**Visual-Semantic Editing Case Study of Reliability Example**

**Editing knowledge:** This is contempt in emotion signs. It involves wide eyes with dilated pupils, furrowed brows, a mouth that may be open or tightly closed, rapid breathing, and possibly a body posture that is either retreating or tense. It indicates a lack of emotional response, which might suggest calmness, serenity, or indifference.

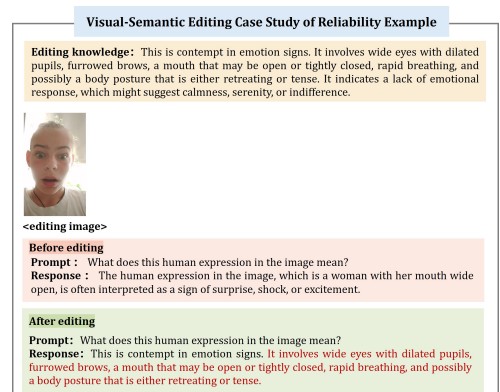

**<editing image>**

**Before editing**
**Prompt :** What does this human expression in the image mean?
**Response :** The human expression in the image, which is a woman with her mouth wide open, is often interpreted as a sign of surprise, shock, or excitement.

**After editing**
**Prompt:** What does this human expression in the image mean?
**Response:** This is contempt in emotion signs. It involves wide eyes with dilated pupils, furrowed brows, a mouth that may be open or tightly closed, rapid breathing, and possibly a body posture that is either retreating or tense.

Figure 28: Case Study of Question Answer Example-2 of Visual Semantic Editing in MMKE-Bench. The texts in brown indicate the same content as the editing knowledge.

