# OpenReview forum: "MMKE-Bench: A Multimodal Editing Benchmark for Diverse Visual Knowledge"
_ICLR.cc/2025/Conference — ICLR 2025 Poster_

### Official Review · Reviewer_PUQh · 2024-10-31

**Soundness:** 2
**Presentation:** 2
**Contribution:** 2
**Rating:** 6
**Confidence:** 4

**Summary:**

This research investigates knowledge editing within large multimodal models, introducing a novel benchmark for evaluation. It defines three specific editing tasks: modifying visual entities, altering visual semantics, and implementing user-specific adjustments, all using natural language as the input format. The benchmark is validated through an assessment of five different editing techniques across three large multimodal models, effectively confirming the dataset's applicability.

**Strengths:**

Utilizing free-form natural language as input for knowledge editing tasks stands out as a key advantage, offering greater flexibility and adaptability to various contexts. The data collection process is robust and comprehensive, and the benchmark supports both individual and sequential editing tasks.

**Weaknesses:**

The training dataset size seems to be small, which might not be enough for mend/ serac that need training. This could potentially lead to lower performance of such methods.

There is potential to enhance the T-Loc test. The current results, which are close to 100, indicate that the randomly selected T-Loc questions may be relatively simple for existing methods. Strengthening the evaluation could involve introducing more challenging cases. For instance, given that GPT was used in data collection, generating analogous but more difficult test cases and validating them could increase the test's effectiveness and robustness.

The division of the three tasks lacks clarity. Both visual entity editing and visual semantic editing require image comprehension and retention of the target knowledge provided in the text. While they may involve different data sources, they essentially fall under the same task, visual question answering. Similarly, user-specific editing could be seen as an application scenario, but it still aligns with the core task of VQA, making no real difference in “task type.”

Additionally, I question about if this user-specific editing approach is practical. In real-world applications, creating personalized databases for each user might be more efficient and effective than modifying the model or embedded knowledge directly.

And the task generalization might not be persuasive using only one case. If you want to prove this, more comprehensive experiments and evaluations should be conducted.

**Questions:**

As mentioned in the weaknesses.

From figure 1, there is no T-Gen test in MMKE-Bench, and you add a T-Rel test. Why do you make this change?

Additionally, how do you adapt each method to the lmm editing?

Figure 4: why I-Gen not evaluated for MMEdit?

Section 5.2.2 requires additional clarification on the terms “user number” and “allowed maximum items.” Further, what about the mend method in your sequential editing setting? And are the gaps substantial enough for the test? In LLM knowledge editing studies and the referenced vlkeb work, this gap can be larger, reaching 100 or more.

The analysis in line 425-426 only explains possible reason for serac, leaving out mend.

Line 450, “visual knowledge” can be ambiguous, and the reliability seems to be not lower as stated.

Line 466, how do you draw the conclusion “parameter-based methods are better at applying edited knowledge to new contexts”?


Writing issue:

double quotes in line 324-327 and elsewhere: please pay attention to how to type correct left quotation mark in latex.

Figure 1: what is the difference between original and editing knowledge of user-specific editing? They are same that lead to confusion, which can be improved for btter clarity.

Figure 2: Should the "G-Rel." change to “T-Rel”? And the two question in the last box (bottom right) are same. And placing “hint” there can be confusing, as if the hint itself is part of the input.

Line 355: you use “MLLMs” here, but in other places, you use LMMs

Table 2: "Entity Entity Editing"

Table 6: missing decimal point in SERAC results, GAP 3 T-Rel column

**Details Of Ethics Concerns:**

No Ethics Concerns

---

> ### Author Response · Authors · 2024-11-21
> **Resopnse to Reviewer PUQh (1/3 parts)**
>
> We sincerely thank you for your time and constructive comments. Below, we provide detailed replies to your comments and hope we can resolve your major concerns.
>
> ---
>
> > W1: The training dataset size seems to be small, which might not be enough for mend/ serac that needs training. This could potentially lead to lower performance of such methods.
>
>
> On the one hand, we aim to demonstrate that our benchmark is comparable in magnitude to existing benchmarks, such as VLKEB and MIKE. On the other hand, to illustrate the impact of training data, we have plotted the training loss of SERAC and MEND in the appendix **(Figures 13–14, Lines 1355–1399)** of the revised manuscript. As shown, the loss consistently decreases during the training process and eventually converges to a small value, indicating that the training data is sufficient for effectively training the model.
>
>
> ---
> > W2: There is potential to enhance the T-Loc test.
>
>
> As for T-LOC, some methods, such as FT-Alignment, achieve nearly 100% accuracy, which can be attributed to the nature of their editing approach. Specifically, FT-Alignment updates only the projection model between the vision and language models. Consequently, for a given text-only question, the LLM's output remains almost identical before and after editing, resulting in a T-LOC close to 100%. Furthermore, to better understand the impact of question complexity on T-LOC, we generated more challenging T-LOC questions using the LLM on visual semantic editing. We provided the original knowledge **(e.g., "This is a fist in a life gesture. The thumb and pinky finger are extended while the other fingers are clenched into a fist. It signifies strength or readiness to strike.")** and item type **(e.g., "life gesture")** to the LLM, prompting it to generate type-related questions **(e.g., "What does it mean when someone waves their hand up and down enthusiastically?")**. We conducted experiments using visual semantic editing data, and the results are summarized in this table. As shown, the performance of most editing methods decreases with harder T-LOC questions, whereas FT-Alignment and SERAC remain nearly unaffected. This further demonstrates that both methods can consistently achieve high T-LOC across different question types.
>
>
> | **Model**        | **Method**     | **T-Loc (new)** | **T-Loc (origin)** |
> |:-----------------:|:--------------:|:---------------:|:------------------:|
> |                   | FT-LLM         | 59.21           | 79.62              |
> |                   | FT-Alignment   | 100.00          | 100.00             |
> | **LLaVA-1.5**     | IKE            | 50.74           | 61.10              |
> |                   | SERAC          | 99.97           | 99.99              |
> |                   | MEND           | 95.65           | 98.15              |
> |                   | KE             | 63.53           | 71.39              |
>
>
>
> ---
> > W3: The division of the three tasks lacks clarity.
>
>
>
> We have included a figure in the appendix **(Figure 12, Line 1322)** of the revised manuscript to illustrate the differences between visual entity editing and semantic editing. In summary, visual entity editing focuses on entity recognition, specifically identifying who or what the entity is, whereas semantic editing emphasizes understanding semantic behaviors, such as gestures, emotions, and actions, without depending on the identity of the entity. Additionally, user-specific editing is designed to incorporate user-related information into the models.
>
>
> ---
> > W4: Additionally, I question about if this user-specific editing approach is practical.
>
>
> We acknowledge that adopting an external memory or database is a viable approach to achieving user-specific applications. However, we emphasize that the need for a user-specific scenario remains realistic regardless of the method employed. Moreover, as the volume of user-specified data increases, the storage requirements expand, and the retrieval performance from memory diminishes, potentially degrading performance in real-world applications.   Therefore, we believe that both adopting external memory and injecting user-specific knowledge into the model are effective solutions for building personalized models in real-world scenarios, each with its own advantages and limitations. Consequently, we propose exploring knowledge editing as an alternative approach to achieve this objective. Finally, it is worth noting that some knowledge editing methods, such as SERAC, also utilize memory-based strategies, which are already incorporated into the benchmark evaluation.

---

> > ### Author Response · Authors · 2024-11-21
> > **Resopnse to Reviewer PUQh (2/3 parts)**
> >
> > ---
> > > W5: And the task generalization might not be persuasive using only one case. If you want to prove this, more comprehensive experiments and evaluations should be conducted.
> >
> >
> >
> > We have included additional task generalization cases in the appendix **(Figures 39–44, Lines 1760–1830)** to provide readers with a clearer understanding of the editing results.
> >
> >
> >
> > ---
> > > Q1: From Figure 1, there is no T-Gen test in MMKE-Bench, and you add a T-Rel test. Why do you make this change?
> >
> >
> > 1) The reason T-GEN is not included in our benchmark is that it is unnecessary for this particular evaluation. In triplet-based evaluations, the reliability question is identical to the training question, necessitating a text generalization question that conveys the same meaning as the original question but with different phrasing. However, in our benchmark, the training question is distinct from the text reliability question, making it unnecessary to rewrite the text reliability question.
> > 2) The rationale for using T-rel as the evaluation metric is that the descriptions in our benchmark encompass multiple facts, and T-rel effectively measures how well these edited facts are learned in a text-only format.
> >
> >
> >
> >
> > ---
> > > Q2: Additionally, how do you adapt each method to the lmm editing?
> >
> >
> > We follow the implementation of MMedit and VLKEB to adapt each method, with VLKEB being built upon MMedit. For FT-LLM, FT-Alignment, KE, and MEND, we update the LLM module within a large multi-modal model. For SERAC, we refine the scope classifier and counterfactual model during training. For IKE, we incorporate the current editing examples alongside selected examples as in-context examples.
> >
> >
> >
> > - **MMedit**: [https://github.com/zjunlp/EasyEdit](https://github.com/zjunlp/EasyEdit)
> > - **VLKEB**: [https://github.com/VLKEB/VLKEB/tree/main](https://github.com/VLKEB/VLKEB/tree/main)
> >
> >
> > ---
> > > Q3: Figure 4: why I-Gen not evaluated for MMEdit?
> >
> > We have re-examined the MMedit results and identified the I-Gen values in Figure 5. Since explicit numerical values are not provided in the figure, we approximate the I-Gen results to be 98.5%. Accordingly, we have re-plotted Figure 4 in the revised manuscript and replaced the original version. As demonstrated, the conclusion remains consistent, reaffirming that our benchmark poses greater challenges compared to previous ones.
> >
> >
> >
> >
> > ---
> > > Q4: Section 5.2.2 requires additional clarification on the terms “user number” and “allowed maximum items.” Further, what about the mend method in your sequential editing setting? And are the gaps substantial enough for the test? In LLM knowledge editing studies and the referenced VLKEB work, this gap can be larger, reaching 100 or more.
> >
> >
> > The terms "user num" and "gap" indeed have different meanings, and we have added a figure in the appendix **(Figure 11, Line 1297)** to illustrate the difference. In summary, the "gap" refers to the number of editing examples in a sequence, while the "user num" represents the number of users in a sequence, where each user may have a different amount of user-specific data (at least one and at most the "allowed maximum items"). The "allowed maximum items" refers to the maximum amount of data for a single user, which is 9 in our benchmark.
> >
> > In addition, we follow the implementation of VLKEB and have not tested MEND in our benchmark. Lastly, we conducted experiments to analyze how performance changes with a larger gap in the visual semantic editing dataset. The results, shown in **Table 14 (Line 1204)** in the appendix, indicate that as the gap increases, the performance of FT-LLM and FT-Alignment methods decreases, while the SERAC method shows little change.
> >
> > ---
> > > Q5: The analysis in lines 425-426 only explains the possible reason for serac, leaving out mend.
> >
> >
> > We apologize for any confusion caused by our previous statements. SERAC utilizes a memory-based approach, whereas MEND employs a parameter-based strategy. SERAC's effectiveness may stem from its utilization of in-context examples, while the success of MEND could be due to minimal modifications to the original model parameters.
> >
> > ---
> > > Q6: Line 450, “visual knowledge” can be ambiguous, and the reliability seems to be not lower as stated.
> >
> > We apologize for the oversight in line 450, where 'visual knowledge' should have been specified as 'visual semantic knowledge'. Regarding portability, the results indicate that the results of visual semantic editing and user-specific editing are lower than those of visual entity editing.

---

> > > ### Author Response · Authors · 2024-11-21
> > > **Resopnse to Reviewer PUQh (3/3 parts)**
> > >
> > > ---
> > > > Q7: Line 466, how do you draw the conclusion that “parameter-based methods are better at applying edited knowledge to new contexts”?
> > >
> > > For this conclusion, we primarily concentrate on the portability results presented in Tables 3-5. It is evident that KE achieves the highest performance in five out of nine tasks and secures the second-best position in four out of the nine tasks. Consequently, we believe that KE is the most effective method for applying editing knowledge to new or related content.  This conclusion is drawn based on KE being a parameter-based method.
> > >
> > > ---
> > > > Q8: Some Writing issues in this paper
> > >
> > >
> > > Thank you for your insightful comments on our manuscript. Regarding Figure 1, we have incorporated the phrase "original knowledge = editing knowledge" in the section on user-specific editing to enhance clarity for our readers. For Figure 2, the term "hint" has been substituted with "explanation" to eliminate ambiguity. Additionally, we have addressed and resolved all other concerns raised in the revision of our paper.

---

> ### Author Response · Authors · 2024-11-25
> **Looking forward to your feedback**
>
> Dear Reviewer PUQh,
>
> We sincerely appreciate your time and effort in reviewing this work. We understand that your schedule may be quite busy. As the authors-reviewer discussion phase draws close, we kindly request your attention to our responses. We aim to gain insights into whether our responses effectively address your concerns and to ascertain if there are any additional questions or points you would like to discuss. We also hope that if you are satisfied with our answers, you may consider adjusting your score accordingly. We look forward to the opportunity to discuss this further with you. Thank you for your thoughtful consideration.

---

> ### Comment · Reviewer_PUQh · 2024-11-25
> **About task generalization**
>
> Thank you for your response. While you have clarified some of my questions, I still have doubts about task generalization. You mentioned the image captioning task; however, in Fig. 6, the example provided is a QA task, where a question is asked, and an answer is given. This is distinct from image captioning, which focuses solely on describing the image.
>
> In the new examples you provided, the input images are identical to the editing image, and the output texts are the same as the editing knowledge. While these examples demonstrate reliability, they do not necessarily support task generalization.
>
> If the benchmark is primarily designed as a QA task, it may not be appropriate to claim task generalization—especially since the examples given are limited and lack a comprehensive convincing evaluation of image captioning. Those examples can serve as case studies, and there may be no need to discuss such task generalization.
>
> P.S. Your revision introduces typos in line 459: "MimiGPT4." It might be worthwhile to check for other issues elsewhere.

---

> > ### Author Response · Authors · 2024-11-25
> > **Response to reviewer PUQh**
> >
> > Dear Reviewer PUQh,
> >
> > Thank you for your thoughtful response. We are pleased to hear that most of your questions have been addressed. Regarding task generalization, we appreciate your suggestion and agree that the question in Fig. 6 aligns more closely with a VQA task format. As you recommended, it would be more appropriate to refer to the examples in Fig. 5, Fig. 6, and the additional examples in the appendix as case studies. We have updated the paper accordingly, changing "task generalization" to "case study." The revised text is highlighted in red for your convenience.
> >
> > Additionally, thank you for pointing out the typographical errors. We have thoroughly reviewed the paper and made the necessary corrections.
> >
> > Best regards.

---

> > > ### Author Response · Authors · 2024-11-27
> > > **Request further feedback**
> > >
> > > Dear Reviewer PUQh,
> > >
> > > Thank you for your valuable feedback during the rebuttal period. We have carefully addressed your comments and made the necessary revisions to the paper.
> > >
> > > If there are any remaining concerns or questions that have not yet been fully resolved, please let us know, and we would be happy to clarify further.
> > >
> > > If all your concerns have been addressed, we would kindly ask you to consider raising your rating based on the changes made.
> > >
> > > We appreciate your time and effort in reviewing our paper and look forward to your final thoughts.
> > >
> > > Best regards

---

### Official Review · Reviewer_Bskj · 2024-11-01

**Soundness:** 2
**Presentation:** 3
**Contribution:** 2
**Rating:** 5
**Confidence:** 4

**Summary:**

This work focuses on knowledge editing in large multimodal models and introduces a new benchmark. It presents three distinct types of editing tasks: visual entity editing, visual semantic editing, and user-specific editing, using free-form natural language as input data for these edits. The benchmark also diverges from previous versions by removing the T-Gen test and adding a T-Rel test. The evaluation includes five editing methods across three large multimodal models (LMMs), effectively validating the benchmark's dataset.

**Strengths:**

The use of free-form natural language as input for knowledge editing tasks is a notable strength, enhancing flexibility and making the approach adaptable. The clarity of the writing aids comprehension, and the experimental setup is well-documented. Additionally, the benchmark spans diverse data sources and entity types, allowing for broad applicability across different tasks.

**Weaknesses:**

There is some overlap between visual entity editing and visual semantic editing, as both tasks involve understanding image content, which could blur the distinction between these editing types. Additionally, the user-specific editing scenario may lack practicality. In real-world applications, database or memory-based search might be more effective than training user-specific information for each user to achieve personalization in LLMs or LMMs.
Regarding the T-Loc test, there’s room for improvement. The results are near 100, suggest that the randomly sampled T-Loc questions are relatively easy for methods such as SERAC, FT-Alignment, and MEND. Introducing more challenging cases could enhance the evaluation’s robustness. For instance, collect similar but harder test cases, either through web crawling or using LLM-generated content (e.g., from GPT) and verify them. This could improve the test’s effectiveness.

**Questions:**

As mentioned in the weaknesses, the similarity of editing visual entity and semantic editing, the user-specific editing scenario and T-Loc test warrant further exploration.

Given the modest amount of training data, is the dataset volume adequate for methods that require training, like serac or mend? Additionally, how is the IKE method adapted to LMM editing? Which examples are used in context? Is target knowledge incorporated within the context, and do you include image-based examples in this context?

In Section 5.2.1, there is a claim that “5) Modern LMMs excel in producing and applying edited knowledge. For reliability, generalization, and portability evaluations, LLaVA-1.5 outperforms BLIP-2 and MiniGPT-4. This improved performance can be attributed to its larger model size and better instruction-following capability.” This raises two questions: First, is LLaVA-1.5 indeed larger than MiniGPT-4, as both use a 7B LLM, and MiniGPT-4's vision encoder appears larger? Second, this statement is not directly related to the benchmark’s core focus, which is to compare editing methods rather than models.

In Section 5.2.2, further clarification is needed regarding the meaning of “user number” and “allowed maximum items.” Additionally, what is the precise gap between editing and testing in user-specific editing, and why is the gap in the two visual editing tasks similar (1, 3, 6, 10) without larger gap?

Typo:
Figure 2, bottom-right: "G-Rel."
Table 2: "Entity Entity Editing"
Table 6: SERAC GAP 3 T-Rel, missing decimal point

---

> ### Author Response · Authors · 2024-11-21
> **Resopnse to Reviewer Bskj (1/2 parts)**
>
> We sincerely thank you for your time and constructive comments. Below, we provide detailed replies to your comments and hope we can resolve your major concerns.
>
> ---
>
> > W1&Q1: There is some overlap between visual entity editing and visual semantic editing, as both tasks involve understanding image content, which could blur the distinction between these editing types.
>
>
> We have included a new figure in the appendix **(Figure 12, Line 1322)** of the revision to illustrate the differences between visual entity editing and semantic editing. In summary, visual entity editing focuses on entity recognition, specifically identifying who or what the entity is, while semantic editing centers on understanding semantic behavior, such as gestures, emotions, and actions, without relying on the identification of the entity.
>
>
>
> ---
>
>
> > W2&Q1: Additionally, the user-specific editing scenario may lack practicality. In real-world applications, database or memory-based search might be more effective than training user-specific information for each user to achieve personalization in LLMs or LMMs.
>
>
> We agree that adopting an external memory or database represents another viable approach to achieving user-specific applications. However, we would like to emphasize that the need for user-specific scenarios remains both realistic and essential for users, irrespective of the specific method employed. Furthermore, as the amount of user-specified data increases, the required storage space would grow, and the retrieval performance from memory would likely degrade, potentially negatively impacting performance in real-world applications. Therefore, we believe that both adopting additional memory and directly injecting user-specific knowledge into the model are viable solutions for achieving personalized models in real-world scenarios, with each approach having its respective strengths and weaknesses. In light of this, we propose exploring knowledge editing as an alternative method for personalization. Finally, it is worth noting that some knowledge editing methods, such as SERAC, also integrate memory-based strategies, meaning that memory-based approaches have already been partially considered in the benchmark evaluations.
>
>
>
> ---
>
> > W3&Q1: Regarding the T-Loc test, there’s room for improvement. The results are near 100, suggesting that the randomly sampled T-Loc questions are relatively easy for methods such as SERAC, FT-Alignment, and MEND. Introducing more challenging cases could enhance the evaluation’s robustness. For instance, collect similar but harder test cases, either through web crawling or using LLM-generated content (e.g., from GPT), and verify them. This could improve the test’s effectiveness.
>
>
> Regarding T-LOC, methods such as FT-Alignment achieve nearly 100%, which is attributed to their editing mechanisms. Specifically, FT-Alignment only updates the projection model between the vision and language models, without modifying the LLM itself. As a result, for a given text-only question, the outputs generated by the LLM before and after editing are nearly identical, leading to a T-LOC close to 100%. To better understand the impact of questions on T-LOC, we generated harder T-LOC questions using the LLM for visual semantic editing. We provided the original knowledge **(e.g., "This is a fist in a life gesture. The thumb and pinky finger are extended while the other fingers are clenched into a fist. It signifies strength or readiness to strike.")** and item type **(e.g., "life gesture")** to the LLM and instructed it to generate type-related questions **(e.g., "What does it mean when someone waves their hand up and down enthusiastically?")**. Experiments were conducted on visual semantic editing data, and the results are shown in the table. As the results indicate, most editing methods perform worse under harder T-LOC questions, while FT-Alignment and SERAC remain nearly unaffected. This demonstrates that both methods can achieve consistently high T-LOC performance across different types of questions.
>
>
>
> | **Model**        | **Method**     | **T-Loc (new)** | **T-Loc (origin)** |
> |:-----------------:|:--------------:|:---------------:|:------------------:|
> |                   | FT-LLM         | 59.21           | 79.62              |
> |                   | FT-Alignment   | 100.00          | 100.00             |
> | **LLaVA-1.5**     | IKE            | 50.74           | 61.10              |
> |                   | SERAC          | 99.97           | 99.99              |
> |                   | MEND           | 95.65           | 98.15              |
> |                   | KE             | 63.53           | 71.39              |

---

> > ### Author Response · Authors · 2024-11-21
> > **Resopnse to Reviewer Bskj (2/2 parts)**
> >
> > ---
> > > Q2: Given the modest amount of training data, is the dataset volume adequate for methods that require training, like SERAC or MEND? Additionally, how is the IKE method adapted to LMM editing? Which examples are used in context? Is target knowledge incorporated within the context, and do you include image-based examples in this context?
> >
> >
> >
> > 1) On the one hand, we aim to demonstrate that our benchmark is of a similar order of magnitude to existing benchmarks, such as VLKEB and MIKE. On the other hand, in order to illustrate the impact of training data, we present the training loss of SERAC and MEND in the appendix **(Figure 13–14, Lines 1355–1399)** of the revision. As shown, during training, the loss continuously decreases and ultimately converges to a small value, suggesting that the training data is sufficient for model training.
> >
> > 2) Regarding the IKE method, we adhere to its original implementation as described in MMedit and VLKEB. The most similar examples, calculated using cosine similarity, are selected as in-context examples, with the target knowledge integrated into the context. For images, we follow the approach of MMedit and VLKEB, excluding images from the context.
> >
> >
> >
> > **MMedit**: [https://github.com/zjunlp/EasyEdit](https://github.com/zjunlp/EasyEdit)
> >
> > **VLKEB**: [https://github.com/VLKEB/VLKEB/tree/main](https://github.com/VLKEB/VLKEB/tree/main)
> >
> > ---
> > > Q3: This raises two questions: 1) First, is LLaVA-1.5 indeed larger than MiniGPT4, as both use a 7B LLM, and MiniGPT4's vision encoder appears larger? 2) Second, this statement is not directly related to the benchmark’s core focus, which is to compare editing methods rather than models.
> >
> >
> > 1) We apologize for the confusion caused by the unclear statement. The LLaVA 1.5 model is of a similar size to MiniGPT4, and its superior performance compared to MiniGPT4 may be attributed to a more effective visual model design and a better instruction tuning strategy. Comparied with Blip2, its enhanced performance may be the result of a larger model size and a more robust training strategy. We sincerely apologize once again for the confusion caused by combining these points in a single sentence, and we have now clarified each of them separately in the revision.
> > 2) Your understanding is accurate. This analysis compares editing methods. We included this comparison to assist readers in better understanding the factors that may influence knowledge editing performance.
> >
> >
> > ---
> > > Q4: 1) In Section 5.2.2, further clarification is needed regarding the meaning of “user number” and “allowed maximum items.” 2) Additionally, what is the precise gap between editing and testing in user-specific editing, and 3) why is the gap in the two visual editing tasks similar (1, 3, 6, 10) without a larger gap?
> >
> >
> > - 1&2) The terms "user num" and "gap" indeed have distinct meanings, and we have included a figure in the appendix **(Figure 11, Line 1297)** to clearly illustrate this distinction. In summary, the "gap" denotes the number of editing examples within a sequence, while the "user num" indicates the number of users within a sequence, with each user potentially having a different amount of user-specific data (ranging from at least one item to at most the "allowed maximum items"). The "allowed maximum items" refers to the maximum amount of data assigned to a single user, which is set to 9 in our benchmark.
> > - 3）Additionally, we performed experiments to investigate how performance varies with a larger gap on the visual semantic editing dataset. The results, provided in **Table 14 (Line 1204)** in the appendix, indicate that as the gap increases, the performance of FT-LLM and FT-Alignment methods declines, whereas the SERAC method demonstrates minimal variation.
> >
> > ---
> > > Q5: Some typos in this paper.
> >
> > Thanks for pointing out the typos in our paper, we have corrected them in the revised version.

---

> ### Author Response · Authors · 2024-11-25
> **Looking forward to your feedback**
>
> Dear Reviewer Bskj,
>
> We sincerely appreciate your time and effort in reviewing this work. We understand that your schedule may be quite busy. As the authors-reviewer discussion phase draws close, we kindly request your attention to our responses. We aim to gain insights into whether our responses effectively address your concerns and to ascertain if there are any additional questions or points you would like to discuss. We also hope that if you are satisfied with our answers, you may consider adjusting your score accordingly. We look forward to the opportunity to discuss this further with you. Thank you for your thoughtful consideration.

---

> ### Author Response · Authors · 2024-12-01
> **Looking forward to your feedback**
>
> Dear reviewer Bskj,
>
> Thanks for your reviews of our work. As the rebuttal deadline is apporaching, we would like to discuss the content with you further and hope to receive your response to the manuscript.
>
> Additionally, if you find that the overall quality of the manuscript has improved after revision, we kindly ask you to consider adjusting the rating score accordingly.
>
> Looking forward to your feedback, thank you!
>
> Best wishes.

---

### Official Review · Reviewer_5HUG · 2024-11-04

**Soundness:** 2
**Presentation:** 2
**Contribution:** 2
**Rating:** 6
**Confidence:** 3

**Summary:**

This paper introduces MMKE-Bench, a multimodal knowledge editing benchmark dataset aimed at addressing the current gap in resources for evaluating large multimodal models (LMMs) in knowledge editing tasks. MMKE-Bench is largely constructed using counterfactual samples to enhance the robustness of model evaluation and to examine their ability to perform knowledge editing across varied and challenging scenarios. The dataset includes three types of tasks: visual entity editing, visual semantic editing, and user-specific knowledge editing. Each task is represented through free-form natural language descriptions combined with images, generated by large language models (LLMs) and verified through human annotation to ensure consistency. The paper evaluates several state-of-the-art multimodal models, including BLIP-2, MiniGPT-4, and LLaVA-1.5, providing insights into the strengths and limitations of current models across different knowledge editing tasks.

**Strengths:**

(1) Diverse Task Setup: MMKE-Bench covers three distinct knowledge editing tasks, from entity-level editing to more complex user-specific knowledge editing. This provides a comprehensive tool for evaluating multimodal models' ability to update knowledge and handle personalized information.

(2) Free-Form Natural Language Descriptions: Unlike traditional triple-based representations, this benchmark uses natural language descriptions to represent knowledge items, enabling models to engage in editing tasks in a more realistic scenario. The free-form descriptions combined with image data make the tasks closer to the complexity of real-world semantics.

(3) Extensive Evaluation Using State-of-the-Art Multimodal Models: The paper evaluates MMKE-Bench on prominent multimodal models such as BLIP-2, MiniGPT-4, and LLaVA-1.5. These models represent the cutting edge in the multimodal field, making the experimental results widely relevant and reflective of current model capabilities in knowledge editing tasks.

(4) Systematic Experimental Analysis: The paper provides a thorough evaluation of various existing knowledge editing methods, including FT-LLM, KE, MEND, SERAC, and IKE, offering a comprehensive performance baseline for each task type. This analysis provides valuable insights into how different methods perform across the tasks presented in MMKE-Bench.

**Weaknesses:**

(1) Figures 1 and 2 Could Benefit from Improved Clarity and Accessibility: Figures 1 and 2 could be refined to enhance clarity and accessibility for a broader audience. In Figure 1, the example used is soccer-related, which may not be immediately understandable to readers unfamiliar with the sport. A more universally recognizable example, such as common objects or activities, could make the data construction process clearer. For Figure 2, the visual design could better distinguish the four dataset construction steps; currently, it’s difficult to determine which modules belong to each step. Using distinct colors, numbering, or borders to separate steps would help readers follow the process more intuitively.

(2) Limitations of Counterfactual Data for Real-World Applications: The counterfactual approach used to construct MMKE-Bench may lead to a distribution that differs from real-world data. While counterfactual samples aid in testing robustness, they represent hypothetical rather than naturally occurring scenarios. As a result, models fine-tuned on this dataset might learn patterns specific to these counterfactual cases, potentially reducing their effectiveness in real-world knowledge editing tasks. A comparison of model performance on both counterfactual and real-world data would provide valuable insights.

(3) Lack of Empirical Analysis on the Impact of Human Verification: The paper mentions that human verification was conducted to improve the consistency of LLM-generated descriptions, but it does not provide examples or quantitative comparisons to illustrate the impact of this verification process. Including a before-and-after analysis would strengthen the case for the added value of human verification.

**Questions:**

(1) Would it be more beneficial to construct knowledge editing samples using incremental, real-world updates? Although challenging, creating samples based on real-world updates, such as player transfers or recent match results sourced from news articles, could enhance dataset relevance. Using real, evolving information would help models learn to handle dynamic knowledge updates and might improve their applicability in practical knowledge editing scenarios.

(2) What impact did human verification have on the LLM-generated descriptions? An analysis or examples of descriptions before and after human verification would help demonstrate how much human intervention improved the dataset's consistency and accuracy.

(3) Is there a comparison of Visual Entity Editing data quality between MMKE-Bench and prior datasets (e.g., MC-MKE, VLKEB)? While Table 1 highlights MMKE-Bench’s increased task diversity with the addition of Visual Semantic Editing and User-Specific Editing, it would be useful to understand how MMKE-Bench compares to previous datasets in terms of data quality for Visual Entity Editing. Such a comparison could clarify whether MMKE-Bench offers improvements in data quality alongside task diversity.

(4) For additional questions, please refer to the Weaknesses section.

---

> ### Author Response · Authors · 2024-11-21
> **Resopnse to Reviewer 5HUG (1/2 parts)**
>
> We sincerely thank you for your time and constructive comments. Below, we provide detailed replies to your comments and hope we can resolve your major concerns.
>
> > W1: Figures 1 and 2 Could Benefit from Improved Clarity and Accessibility
>
> We have improved Figure 1 and Figure 2 in the revised version to help readers better understand our work:
>
> - Figure 1: The example of a soccer referee gesture has been replaced with a more relatable example of a common gesture from everyday life, making it more familiar to most readers.
> - Figure 2: The original figure has been updated by adding four bounding boxes and serial numbers to clarify the data generation steps. Additionally, typos have been corrected to ensure greater accuracy and clarity.
>
> ---
> > W2: Limitations of Counterfactual Data for Real-World Applications
>
> We acknowledge that counterfactual data may differ in distribution from real-world data. However, collecting real-world data, particularly for capturing genuine changes in visual semantic editing, is challenging. As a result, we adopt the widely used counterfactual data generation strategy in LLM and LMM research [1–3] to construct our benchmark.
>
> Regarding experiments on real-world data, to the best of our knowledge, there are currently no datasets or benchmarks based on real-world data specifically designed for multimodal knowledge editing. This lack of resources makes it infeasible to conduct such experiments at this time.
>
> That said, we believe that collecting real-world data for multimodal knowledge editing is a valuable research direction and are considering exploring this avenue in the future.
>
> ---
> > W3: Lack of Empirical Analysis on the Impact of Human Verification
>
> Human verification is integrated into several steps of the benchmark construction process, including candidate item generation, image filtering, counterfactual editing verification, evaluation question generation (remove or rewrite), and overall quality checks.
>
> Among these steps, image filtering requires significant human effort, as images downloaded from Google often include noisy or irrelevant content, and some images must be manually extracted from videos. After this human filtering process, the overall quality of the images is notably improved.
>
> For counterfactual editing verification, we found that the majority of the generated counterfactual content by LLM based on original knowledge was of high quality, with a success rate of nearly 100%. Consequently, no additional human verification was required for this step.
>
> Regarding evaluation question generation, we used LLMs to generate questions, followed by human verification. Compared to counterfactual editing, the quality of LLM-generated questions was lower. To provide a quantitative comparison, we evaluated 100 questions generated by the LLM. Among them, 81 questions and their corresponding answers were deemed acceptable, achieving a success rate of 81%. After human verification and rewriting, noisy or incorrect questions were either corrected or removed, further improving the overall quality.
>
> ---
> > Q1: Would it be more beneficial to construct knowledge editing samples using incremental, real-world updates?
>
>
> We agree that constructing real-world editing data for evaluation is a valuable research direction, and we are considering exploring this as part of our ongoing work.
>
> ---
> > Q2: What impact did human verification have on the LLM-generated descriptions?
>
> We have explained the impact of human verification in W3, where both qualitative analysis and quantitative comparisons are provided. We believe this step significantly enhances the quality of our benchmark.
>
> ---
> > Q3: Is there a comparison of Visual Entity Editing data quality between MMKE-Bench and prior datasets (e.g., MC-MKE, VLKEB)?
>
> As explained earlier, the quality of our benchmark is ensured through automated generation steps, which are complemented by extensive human verification, particularly for visual entity editing data.
>
> In terms of comparison, the data collection and question generation steps are similar to those in previous benchmarks. For entity and image collection, we first gather entity items from existing datasets, retrieve corresponding images from Google, and then filter out noisy images through human verification. In contrast, MC-MKE requires additional subject recognition in a sentence. The quality of data collection is primarily enhanced through human verification, with significant human effort invested to ensure high image quality.
>
> Regarding knowledge representation, both MC-MKE and VLKEB use a triplet knowledge format, while our benchmark adopts a natural language-based knowledge representation. Descriptions are sourced from Wikipedia to ensure the quality of the original content, followed by counterfactual editing performed by LLMs.
>
> We believe that both the previous benchmarks and our own achieve a high level of quality.

---

> > ### Author Response · Authors · 2024-11-21
> > **Resopnse to Reviewer 5HUG (2/2 parts)**
> >
> > ---
> > ### References
> > [1] Yunzhi Yao, Peng Wang, Bozhong Tian, Siyuan Cheng, Zhoubo Li, Shumin Deng, Huajun Chen, and Ningyu Zhang. Editing large language models: Problems, methods, and opportunities. Findings of EMNLP, 2023.
> >
> > [2] Siyuan Cheng, Bozhong Tian, Qingbin Liu, Xi Chen, Yongheng Wang, Huajun Chen, and Ningyu Zhang. Can we edit multimodal large language models? In EMNLP, pp. 13877–13888, 2023.
> >
> > [3] Han Huang, Haitian Zhong, Tao Yu, Qiang Liu, Shu Wu, Liang Wang, and Tieniu Tan. Vlkeb: A large vision-language model knowledge editing benchmark. NeurIPS, 2024.

---

> ### Author Response · Authors · 2024-11-25
> **Looking forward to your feedback**
>
> Dear Reviewer  5HUG,
>
> We sincerely appreciate your time and effort in reviewing this work. We understand that your schedule may be quite busy. As the authors-reviewer discussion phase draws close, we kindly request your attention to our responses. We aim to gain insights into whether our responses effectively address your concerns and to ascertain if there are any additional questions or points you would like to discuss. We also hope that if you are satisfied with our answers, you may consider adjusting your score accordingly.
> We look forward to the opportunity to discuss this further with you. Thank you for your thoughtful consideration.

---

> > ### Comment · Reviewer_5HUG · 2024-11-29
> >
> > Thank you for the response, which addressed most of my concerns. I will maintain my score of 6.

---

> ### Author Response · Authors · 2024-11-29
> **Thanks for your feedback**
>
> Dear Reviewer 5HUG,
>
> Thank you for your feedback. Your insights have been invaluable in enhancing our manuscript.
>
> If there are any remaining concerns or questions that have not yet been fully resolved, please let us know, and we would be happy to clarify further.
>
> We appreciate your time and effort in reviewing our paper again.
>
> Best regards

---

### Official Review · Reviewer_1STo · 2024-11-04

**Soundness:** 3
**Presentation:** 3
**Contribution:** 3
**Rating:** 8
**Confidence:** 3

**Summary:**

This paper presents a benchmark for knowledge editing in multimodal large models, specifically targeting knowledge represented in free-form natural language. The benchmark focuses on tasks such as visual entity editing, visual semantic editing, and user-specific editing. The paper provides a detailed description of the data collection and construction process for this benchmark. Various model editing methods were evaluated on this benchmark using models like BLIP-2, MiniGPT-4, and LLaVA 1.5, revealing limitations in existing approaches.

**Strengths:**

1. The benchmark proposed in this paper uses knowledge represented in free-form natural language, making it more applicable to real-world scenarios. In addition to traditional visual entity editing, the benchmark incorporates visual semantic editing and user-specific editing, allowing for a more comprehensive evaluation of model editing capabilities.
2. The paper provides a detailed description of the dataset construction process, offering valuable insights and methodologies for data collection and structuring.
3. The experiments conducted on the benchmark cover a wide range of model editing methods across different types of knowledge editing tasks, resulting in an extensive and insightful evaluation.

**Weaknesses:**

1. I’m not sure if the workload is sufficient. If this were solely a Dataset & Benchmark Track submission, the workload would be appropriate. However, as a long paper submission to ICLR, it may require additional technical contributions. For instance, adding theoretical analysis to explain why existing methods perform poorly in multimodal knowledge editing could provide new perspectives for improving this area. Therefore, it’s recommended to include an in-depth analysis on why current methods underperform in certain aspects relevant to the proposed benchmark. It is suggested to provide related theoretical analysis or technical contributions that you believe would enhance the paper's suitability for ICLR.

2. I didn’t find evidence in the experimental or case study sections demonstrating the necessity of this benchmark. This is not to question the value of your work, but providing examples would strengthen the case. Additionally, clarifying which scenarios would particularly benefit from using this benchmark for evaluation would be helpful. It’s recommended to add more evidence highlighting the ‘differences’ and ‘necessity’ of this benchmark. I suggest you provide comparative analyses with existing benchmarks or specific real-world scenarios where this benchmark would be particularly valuable.

**Questions:**

1. In line 776, “counterfactual editing” is mentioned. According to the description in lines 776-781, modifying certain facts constitutes counterfactual editing. Is this definition accurate?
2. The case study in Section 5.3 could include more analysis. Figure 5 and Figure 6 only present a relatively simple entity editing example (person-related information) and do not showcase complex cases of visual semantic editing or user-specific knowledge editing. It’s recommended to select representative and challenging cases or specific types of cases in visual semantic editing or user-specific knowledge editing.
3. The superiority of IKE has been demonstrated in many previous works. What new perspective does your benchmark provide? Section 5.3 only states that IKE performs better than FT-LLM without explaining why IKE has stronger generalization and transferability. It would be helpful to explain the reasons for the performance differences across methods based on the proposed benchmark. It is suggested to provide a detailed analysis of how IKE's performance on this new benchmark differs from its performance on previous benchmarks, and what these differences reveal about the nature of multimodal knowledge editing tasks.
4. This benchmark is designed to uncover issues that other benchmarks cannot detect. Could you provide some examples or data to demonstrate that your benchmark is indeed more challenging and more valuable for improving models compared to other benchmarks?

---

> ### Author Response · Authors · 2024-11-21
> **Resopnse to Reviewer 1STo (1/2 parts)**
>
> We sincerely thank you for your time and constructive comments. Below, we provide detailed replies to your comments and hope we can resolve your major concerns.
>
> ---
> > W1: I’m not sure if the workload is sufficient.
>
> Firstly, we have selected the Primary Area of Datasets and Benchmarks for our submission to ICLR 2025, aligning with the scope and focus of our work. Therefore, we believe our contribution is well-suited for consideration at the conference.
>
> Secondly, beyond introducing a novel benchmark, we also propose an innovative pipeline for benchmark construction, which we hope will inspire further research in the field. Our findings also underscore the limitations of existing knowledge editing methods, highlighting the need for more advanced approaches to multi-modal knowledge editing.
>
>
> Thirdly, we discuss the experiment results as follows. A major difference between our benchmark and existing ones lies in that the training and evaluation questions are the same as the evaluation question (reliability) as there are multiple editing facts in one sample, However, in our benchmark, the training and evaluation questions differ. Thus, the editing method on our benchmark is more challenging. Furthermore, the evaluation task in our benchmark emphasizes semantic understanding, which is inherently more difficult than the entity recognition focus of the previous benchmarks. Moreover, the evaluation tasks in our benchmark emphasize semantic understanding, which is inherently more complex than the entity recognition tasks emphasized in previous benchmarks. This distinction further underscores the unique contributions of our work.
>
> ---
> > W2: I didn’t find evidence in the experimental or case study sections demonstrating the necessity of this benchmark.
>
> The case studies in our experiments primarily demonstrate the effectiveness of knowledge editing methods. To better highlight the ‘differences’ and ‘necessity’ of our benchmark, we have added additional data examples in the appendix (**Figure 15–26, lines 1404–1560**). Based on these examples, the ‘differences’ and ‘necessity’ can be summarized as follows:
>
> - 1) **Necessity in real-world scenarios**:
> Taking daily hand gestures as an example, the same gesture may convey friendliness in one country but appear unfriendly in another. Therefore, it is crucial to edit knowledge about the meanings of specific hand gestures to avoid cultural conflicts. Similarly, when entering an unfamiliar country or region, we may encounter hand gestures that are unfamiliar to us. Recognizing and adopting these gestures can facilitate better communication. Both scenarios underscore the need for editing methods to improve the recognition and understanding of hand gestures, aligning with the motivation for visual semantic editing.
>
> - 2) **Differences from other benchmarks**:
> Previous multi-modal benchmarks primarily focus on entity-level knowledge editing using a triple-based knowledge representation format. In contrast, our benchmark evaluates complex visual semantic scenarios with a natural language-based knowledge representation. In summary, the key differences between our benchmark and existing ones are: 1) Focus on complex visual semantic editing rather than simple entity-level editing. 2) Adoption of a flexible natural language-based knowledge representation format, enabling richer and more context-aware editing capabilities.
>
> ---
> > Q1: In line 776, “counterfactual editing” is mentioned. According to the description in lines 776-781, modifying certain facts constitutes counterfactual editing. Is this definition accurate?
>
> Your understanding is correct. The counterfactual editing strategy, or the generation of counterfactual editing descriptions, is widely adopted in the generation of knowledge editing datasets for LLM- and LMM-based benchmarks [1–3]. This approach involves modifying factual knowledge into counterfactual content to construct editing datasets. Similarly, in our benchmark, we employ this strategy by editing specific facts into counterfactual content, aligning with the concept of counterfactual editing.

---

> ### Author Response · Authors · 2024-11-21
> **Resopnse to Reviewer 1STo (2/2 parts)**
>
> ---
> > Q2: The case study in Section 5.3 could include more analysis. Figure 5 and Figure 6 only present a relatively simple entity editing example (person-related information) and do not showcase complex cases of visual semantic editing or user-specific knowledge editing. It’s recommended to select representative and challenging cases or specific types of cases in visual semantic editing or user-specific knowledge editing.
>
>
> Firstly, there appears to be a misunderstanding regarding Figure 6. While it is person-related, its primary purpose is to illustrate a case of visual semantic editing. Specifically, the focus is on understanding the meaning of the gesture performed by the basketball referee (visual semantic understanding) rather than identifying the individual performing it (entity recognition).
>
> Secondly, based on your feedback and the suggestion from the reviewer PUQh, we have changed "task generalization" in Fig 6 to "case study of reliability" as they align more with the VQA question format. Besides, we have included an additional case study on editing examples (**Figure 27–38, lines 1579–1766**) and a case study of reliability examples (**Figure 39–44, lines 1760–1830**) in the appendix of the revised version to provide clearer visualizations and enhance the overall understanding.
>
> ---
> > Q3: It is suggested to provide a detailed analysis of how IKE's performance on this new benchmark differs from its performance on previous benchmarks, and what these differences reveal about the nature of multimodal knowledge editing tasks.
>
> We compare IKE's performance with existing benchmarks, such as VLKEB, as follows:
>
>
> - The performance of editing methods on our benchmark aligns with results observed in existing benchmarks, where IKE achieves the best or second-best results in terms of reliability and generality. However, IKE demonstrates significant improvements over FT-LLM in terms of portability for VLKEB across all models (BLIP2, MiniGPT4, and LLaVA). In contrast, on our benchmark, IKE shows limited improvement or even underperforms compared to FT-LLM on BLIP2 and MiniGPT4 but achieves notable improvement on LLaVA 1.5. We attribute this to the complexity of understanding longer natural language descriptions. Modern models, such as LLaVA 1.5, benefit from better training strategies, enabling them to handle such descriptions more effectively, resulting in superior performance. Besides, The results indicate that IKE performs better in complex visual semantic and user-specific editing scenarios in terms of portability, highlighting the challenges in transferring semantic and user-specific editing tasks.
>
>
> - In-context examples play a critical role in enabling models to understand the editing tasks in IKE. Unlike triple-based knowledge formats, the natural language format adopted by our benchmark requires longer input lengths. Consequently, the number of in-context examples in our benchmark is smaller compared to other benchmarks when the maximum input length for the base model is fixed. This distinction further differentiates our benchmark from existing ones.
>
>
> ---
> > Q4: Could you provide some examples or data to demonstrate that your benchmark is indeed more challenging and more valuable for improving models compared to other benchmarks?
>
>
>
>
> We have discussed the differences in both the Introduction and Experiment sections. To better illustrate the comparison, we have included additional data examples from our benchmark in the appendix (**Figure 15–26, lines 1404–1560**) of the revised version and provided further discussion in W2.
>
>
>
> ---
>
> ### References
>
> [1] Yunzhi Yao, Peng Wang, Bozhong Tian, Siyuan Cheng, Zhoubo Li, Shumin Deng, Huajun Chen, and Ningyu Zhang. Editing large language models: Problems, methods, and opportunities. Findings of EMNLP, 2023.
>
> [2] Siyuan Cheng, Bozhong Tian, Qingbin Liu, Xi Chen, Yongheng Wang, Huajun Chen, and Ningyu Zhang. Can we edit multimodal large language models? In EMNLP, pp. 13877–13888, 2023.
>
> [3] Han Huang, Haitian Zhong, Tao Yu, Qiang Liu, Shu Wu, Liang Wang, and Tieniu Tan. Vlkeb: A large vision-language model knowledge editing benchmark. NeurIPS, 2024.

---

> ### Author Response · Authors · 2024-11-25
> **Looking forward to your feedback**
>
> Dear Reviewer 1STo,
>
> We sincerely appreciate your time and effort in reviewing this work. We understand that your schedule may be quite busy. As the authors-reviewer discussion phase draws close, we kindly request your attention to our responses. We aim to gain insights into whether our responses effectively address your concerns and to ascertain if there are any additional questions or points you would like to discuss. We also hope that if you are satisfied with our answers, you may consider adjusting your score accordingly.
> We look forward to the opportunity to discuss this further with you. Thank you for your thoughtful consideration.

---

> > ### Author Response · Authors · 2024-12-01
> > **Looking forward to your feedback**
> >
> > Dear reviewer 1STo,
> >
> > Thanks for your reviews of our work. As the rebuttal deadline is apporaching, we would like to discuss the content with you further and hope to receive your response to the manuscript.
> >
> > Additionally, if you find that the overall quality of the manuscript has improved after revision, we kindly ask you to consider adjusting the rating score accordingly.
> >
> > Looking forward to your feedback, thank you!
> >
> > Best wishes.

---

> ### Comment · Reviewer_1STo · 2024-12-02
>
> I have carefully reviewed your revised version, including both the main text and appendix. I deeply appreciate the detailed explanation of visual semantic editing examples in Figure 1 and the comprehensive examples in Figures 15-44 in the appendix, as these have greatly helped me better understand your work. Your responses have addressed all my concerns.
>
> Additionally, I have a minor question regarding the response to W1, where it states: "Thirdly, we discuss the experiment results as follows. A major difference between our benchmark and existing ones lies in that the training and evaluation questions are the same as the evaluation question (reliability) as there are multiple editing facts in one sample, However, in our benchmark, the training and evaluation questions differ." Could you elaborate on this point? I don't fully understand this distinction, since other benchmarks also focus on reliability, generalization, and portability?

---

> ### Author Response · Authors · 2024-12-02
> **Response to Reviewer 1STo**
>
> Dear Reviewer 1STo,
>
> Thank you for your thoughtful response. We are pleased to hear that all of your previous concerns have been addressed.
>
> Regarding your further question, we would like to clarify the following points. Firstly, there seems to be some confusion in our previous response. What we intended to convey is that:
>
> 1.	In the previous benchmark, the question in the training data (i.e., the editing knowledge in one example) is the same as the evaluation question (i.e., the reliability question), as there is only one editing fact per example.
>
> 2.	In contrast, in our benchmark, the training data question differs from the evaluation question (i.e., the reliability question) because there are multiple edited facts within a single example, due to the long, free-form natural language format.
>
> Besides, we have provided a detailed example that appears in Figure 1 below for clarity.
>
> >**Previous Benchmark**:
>
> >Training data:
>
> >Original Knowledge: {Q: Who is the person in the image? A: Wayne Rooney}
>
> >Editing Knowledge: {Q: Who is the person in the image? A: **Zlatan Ibrahimović**}
>
> (Where **bold** indicates the edited part.)
>
> >Evaluation data:
>
> >Reliability question: {Rel: Who is the person in the image? A: Zlatan Ibrahimović}
>
> As we can see, the training question is identical to the reliability question.
>
> >**Our Benchmark**:
>
> >Training data:
>
> >Original Knowledge: {Q: Give me some important information about the person in the image. A: The person in the image is Zlatan Ibrahimović. He is a Swedish former professional footballer. He began his career at Malmö FF in 1999 and rejoined Milan in 2020.}
>
> >Editing Knowledge: {Q: Give me some important information about the person in the image. A: The person in the image is Zlatan Ibrahimović. He is an **Italian** former professional footballer. He began his career at **Roma** in 1999 and rejoined **Real Madrid** in 2020.}
>
> (Where **bold** indicates the edited part.)
>
> >Evaluation data:
>
> >Reliability: {Rel: Which club did the person in the image rejoin in 2020? A: Real Madrid}
>
> In this case, the training question is generated using a template we created. For visual entity editing, the template is “Give me some important information about the {type} in the image”, where {type} refers to the entity type derived from the Wikipedia API. As we can see, the training question differs from the reliability question, as there are multiple edited facts in a single example.
>
> Additionally, as you pointed out, the overall evaluation criteria are the same as those in the previous benchmark, with a slight difference. T-Gen, which was tested in the previous benchmark, is not tested in MMKE-Bench. This is because it is unnecessary for this particular evaluation. As we explained earlier, in the previous benchmark, the reliability question is identical to the training question, thus, a text generalization question that conveys the same meaning as the original question but with different phrasing is required. However, in our benchmark, the training question is distinct from the reliability question, making the rewriting of the reliability question redundant.
>
>
> We hope this response clarifies the issue. If your concerns have been addressed, we kindly request that you consider adjusting your rating based on the changes made.
>
> Best regards.

---

> > ### Comment · Reviewer_1STo · 2024-12-02
> >
> > Thank you for your clear explanation.
> >
> > Previously, I thought the main distinctions between your work and others were in visual semantic editing and user-specific editing. (I agree with other reviewers that user-specific editing might be solved differently for personalization, but it fits well here as a benchmark for model editing capabilities)
> >
> > However, through your response to this question, I now recognize that expressing and editing knowledge in natural language format is another significant highlight of your benchmark and pipeline.
> >
> > I intend to increase my rating of this work to 8 points.

---

### Author Response · Authors · 2024-11-21
**Summary of Paper Revision**

We sincerely thank all the reviewers for their constructive and thoughtful feedback. We are deeply encouraged by the positive comments that highlight the realism and diversity of our benchmark, the insightful data collection pipeline, the extensive experimental evaluation, the systematic analysis, and the quality of our writing. We greatly appreciate the reviewers' recognition of our work.

In response to the reviewers' comments, we have addressed each question individually. Additionally, we have revised our manuscript based on the feedback, with all changes highlighted in blue. Below, we provide a summary of the revisions:

1. **Clearer Illustration of the Benchmark**. 1) Improved Figure 1 by replacing the original example with a more common data example for visual semantic editing and adding an explanation for user-specific editing. 2) Added Figure 12 in the appendix to show the differences between visual entity editing and visual semantic editing. 3)  Included more data examples from our benchmark in Figures 15–26 in the appendix.

2. **Clearer Benchmark Construction Pipline**. Enhanced Figure 2 with more prominent bounding boxes and serial numbers to clarify the process.

3. **Clearer Experimental Settings and Additional Results**. 1) Added Figure 11 in the appendix to illustrate the experimental setup for sequential editing. 2) Conducted further experiments on sequential editing with larger gaps (e.g., 40, 60, 80), with results presented in Table 14 in the appendix.

4. **Enhanced Experimental Analysis**. 1) Added the training loss curves of MEND and SERAC in Figures 13 and 14 in the appendix. 2) Provided clearer experimental analysis of some editing methods in Section 5.2.1. 3) Improved Figure 4 by including the I-Gen results of MMEdit.

5. **More Case Studies Results**. 1) Added more case study examples in Figures 27–38. 2) As suggested by the reviewer PUQh,  changed "task generalization" in Fig 6 to "case study of reliability", and included an additional case study of reliability examples in Figures 39–44 in the appendix.

6. **Correction of Typos**. Fixed all typos noted by the reviewers.

Thank you once again for your valuable time and effort in reviewing our paper! We hope these revisions address your concerns and improve the clarity and quality of our work.

---

### Meta-Review · Area_Chair_zNbP · 2024-12-18

**Metareview:**

Summary:
The paper introduces MMKE-Bench, a multimodal benchmark evaluating diverse knowledge editing tasks: visual entity editing, visual semantic editing, and user-specific editing. It employs natural language input, enhancing real-world applicability. Experiments with state-of-the-art methods highlight the challenge of visual and user-specific edits, emphasizing current limitations.

Strengths:
1) Novel and comprehensive benchmark for multimodal editing.
2) Robust evaluation across editing methods and models.

Weaknesses
The practicality of user-specific editing scenarios remains unclear.

The benchmark is timely, addresses an important gap, and stimulates progress in multimodal knowledge editing. Revisions effectively addressed key concerns.

**Additional Comments On Reviewer Discussion:**

The reviewers raised concerns about task overlap between visual entity and semantic editing, the practicality of user-specific editing, and the simplicity of T-LOC tests. The authors clarified task distinctions, justified user-specific scenarios, and introduced harder T-LOC cases. The revisions were satisfactory, supporting acceptance due to improved clarity and robustness.

---

### Decision · Program_Chairs · 2025-01-22

Accept (Poster)